# DebrisInterMixing-2.3: a finite volume solver for three-dimensional debris-flow simulations with two calibration parameters - Part 2: Model validation with experiments

Albrecht von Boetticher[1,3], Jens M. Turowski[2], Brian W. McArdell[3], Dieter Rickenmann[3], Marcel Hürlimann[4], Christian Scheidl[5], and James W. Kirchner[1,3]

[1]Department of Environmental Systems Science, Swiss Federal Institute of Technology Zurich ETHZ, CHN H41, 8092 Zürich, Switzerland
[2]Helmholtz-Centre Potsdam GFZ German Research Center for Geosciences, Telegrafenberg, 14473 Potsdam, Germany
[3]Swiss Federal Research Institute WSL, Zürcherstrasse 111, 8903 Birmensdorf, Switzerland
[4]Department of Geotechnical Engineering and Geosciences, Technical University of Catalonia UPC, Jordi Girona, 1-3 (D2), 08034 Barcelona, Spain
[5]Institute of Mountain Risk Engineering, BOKU, Peter-Jordan-Straße 82, 1190 Vienna, Austria

*Correspondence to:* Albrecht v. Boetticher (albrecht.vonboetticher@usys.ethz.ch)

**Abstract.** Here we present validation tests of the fluid dynamic solver presented in von Boetticher et al. (2016), simulating both laboratory-scale and large-scale debris-flow experiments. The new solver combines a Coulomb viscoplastic rheological model with a Herschel-Bulkley model based on material properties and rheological characteristics of the analysed debris flow. For the selected experiments in this study, all necessary material properties were known – the content of sand, clay (including its mineral composition) and gravel as well as the water content and the angle of repose of the gravel. Given these properties, two model parameters are sufficient for calibration, and a range of experiments with different material compositions can be reproduced by the model without recalibration. One calibration parameter, the Herschel-Bulkley exponent, was kept constant for all simulations. The model validation focuses on different case studies illustrating the sensitivity of debris flows to water and clay content, channel curvature, channel roughness and the angle of repose. We characterize the accuracy of the model using experimental observations of flow head positions, front velocities, run-out patterns and basal pressures.

## 1 Introduction

Debris flows are a frequent natural hazard in mountain regions. They consist of a mixture of water, clay, sand and coarser material traveling as a partially or fully fluidized mass through steep channels. The mixture of different materials leads to a complex rheological behavior that is still not well understood. Field observations of debris-flow behavior and rheology are challenging and still rare, and numerical modeling is often the approach of choice when assessment of debris-flow behavior is needed for planning, zoning, and hazard assessment (Scheuner et al., 2011; Christen et al., 2012; Kattel et al., 2016; Mergili et al., 2017). Most models require direct calibration to capture the site-specific behavior. However, reliable calibration data are rare, and laboratory experiments are difficult to upscale to field situations. von Boetticher et al. (2016) recently presented a new

solver that was designed for the simulation of debris flow behavior based on only two free model parameters. In contrast to the common depth-averaged model approaches for debris flow simulation, this model resolves the flow process in three dimensions. Thus the strong coupling between the flow behavior and the channel geometry and basal roughness can be addressed as shown within this work. The model treats the air and fluid phases separately, and derives the properties of the latter by concentration-dependent mixing of a granular material fraction with a fine material suspension. The granular material fraction is modeled as a fluid with Coulomb-viscoplastic rheology (Pudasaini (2012), Domnik et al. (2013)) and the fine material suspension is characterized by a Herschel-Bulkley rheology. The local rheology of the bulk mixture is obtained from the rheological properties of the modeled gravel and the rheology of the fine material suspension, as a linearly weighted average of the corresponding shares in the debris flow material. The rheology of the gravel is defined by its angle of repose and the material properties of the fine material suspension are related to the fractions of different clay minerals and to the water content. The composition is pre-defined, and no dynamic changes of the gravel concentration or of the share of fine material suspension are modeled at this stage. For the dynamic evolution of the solid and fluid concentrations and a separate treatment of velocities per phase we refer to more general models and simulations (Pudasaini, 2012; Mergili et al., 2017).

The objective of this study is to illustrate the model's ability to accurately account for a wide range of flow behaviors without recalibration. The key attributes of the model are its sensitivity to water content, gravel- and clay-fraction and clay-mineralogy on the one hand (also see Haas et al. (2015)), and the interaction between the phase-averaged bulk rheology of the mixture and the complex three-dimensional flow structure on the other.

We first present validation test cases that focus on water content sensitivity in laboratory scale, followed by a model setup to analyze the effect of enhanced free surface elevations due to channel curvature. We then study the model's capability to adapt to basal roughness using large-scale flume experiments. Finally, we illustrate the role of the gravel rheology on the overall simulation results using large-scale experiments with a water-sand-gravel mixture. We discuss limitations of the model set-up based on these simulation results.

## 2   Model concept

The model, as described by von Boetticher et al. (2016), is based on an adaptation of the interMixingFoam solver of the open source finite volume code OpenFOAM (OpenFOAM-Foundation, 2014). We linked the Herschel-Bulkley rheology parameters to the composition of the material mixture and assumed that high contents of fine material such as the interstitial suspension between the gravel grains can damp grain-to-grain collisions. Under this assumption, the gravel can be treated as a Coulomb-viscoplastic fluid with the pressure-dependent rheology model of Domnik et al. (2013). The stable implementation together with the reduction to two free model parameters allows reliable numerical studies of three-dimensional flow processes of debris flows that have high shares of fine material.

The bulk mixture is combined with an air phase by the volume-of-fluid method (Hirt and Nicholsl, 1981) to capture the free surface. In addition to determining typical material parameters (density, water content and relative amounts of gravel and clay), the user is required to input the clay composition (e.g., the fractions of kaolinite and chlorite, illite, montmorillonite;

(Yu et al., 2013)), and $\delta$, the friction angle of the gravel fraction, approximated as its angle of repose. To be in agreement with the experiments of Yu et al. (2013), we consider all particles below 2 mm grain size as part of the interstitial slurry. The two remaining calibration parameters are related to the fine material suspension. One of the two free model parameters, the Herschel-Bulkley exponent $n$, was kept constant and set to 0.34, which was suitable for all simulations presented here. Due to that, the only parameter modified for calibration was $\tau_{00}$, which acts as a multiplication factor for the calculated yield stress of the fine sediment suspension. In case of dense mixtures where the volumetric solid concentration exceeds a threshold of 0.47, the model amplifies $\tau_{00}$ as defined in (Yu et al., 2013).

## 3 Model validation and performance based on selected flume experiments

Three different experimental setups were chosen to illustrate how sensitively the modeled flow and depositional processes react to changes in water and clay content, channel curvature and bed roughness. The first experimental case used for validation is based on flume experiments from Hürlimann et al. (2015), simulating hillslope debris flows on a wide laboratory slope to exclude side-wall effects as suggested by Jop et al. (2008). We use this case study to illustrate to what extent the calibrated model can predict flow behavior with different water contents without recalibration.

The second experimental case used for validation was designed to study the sensitivity of debris flows to channel curvature (Scheidl et al., 2015). The channel had a semi-circular cross-section and was composed of two curves with different radii. The experimental setup focused on surface super-elevation (lateral difference in flow surface elevation in a channel bend), and we consider it as suitable for verifying the modeled rheology of the mixture in channel bends. We mention that effects of curvature were analytically modeled and validated for dry granular flows and flows of mixtures by Pudasaini et al. (2005, 2008) with a depth-averaged approach.

While the two sets of experiments described above were performed with small amounts of debris flow material, over short times and at a laboratory scale, we also tested our model against data from full-scale experiments performed in the USGS experimental debris-flow flume at the H. J. Andrews Experimental Forest, Oregon (Iverson et al., 2010). In these experiments, debris flow material was released into a 2 m wide and about 75 m long flume with 31° inclination, followed by a smooth transition into a planar run-out area with a 2.5° slope in the flow direction. In the initial experiments, the channel was a flat concrete bed; in later experiments, the bed was paved with 1.6 cm high bumps.

The model setup and performance for all three cases are described in more detail in the following sections.

### 3.1 Boundary conditions and release setup

The mobilization of debris flow material in nature is often linked to a change in water content and pore pressure and a corresponding state-dependent transient weakening (Iverson et al., 1997). The focus of this work is not on the release process but on the capability to predict runout distances and impact pressures, and instead of a natural mobilization, the selected experiments released material by opening a head gate. We performed our simulations as sudden dam-break releases (i.e., without a gate) and compared them to simulations including the dynamic motion of the gate. In the first case study the gate was removed ver-

tically in a fast upward motion (Fig. 1), while at the USGS flume the gate, consisting of two wings, was unlocked and pushed open by the material. Although the gate opening had an effect on the formation of the front flow depth (Fig. 2), the difference between simulations with or without the gate vanished during the flow process prior to reaching points of measurements that we used for comparison. A no-slip boundary was applied in all simulations, which is only appropriate in situations with high contents of fine material where viscoplastic behavior dominates. In general, the solid particles exhibit slip and viscous fluid exhibits no-slip along the basal surface. Only through the core assumption of identical velocity between gravel and surrounding fluid due to high drag do such distinct basal boundary conditions reduce to a no-slip condition. OpenFOAM offers partial-slip boundary conditions, however, the definition would only become meaningful together with real two-phase mass flow models as in Pudasaini (2012); Mergili et al. (2017), or as developed by Wardle and Weller (2013) within our 3D framework. The application of a no-slip boundary condition to the USGS flume experiment with a sand-gravel mixture (called SG mixture in the following) on a smooth bed does not fulfill this requirement, and was only chosen in order to have a model setup which is comparable to the sand-gravel mixture with loam (called SGM mixture in the following).

## 3.2 Grid resolutions

In general, we distinguished between channelized flows and flows on a plane in choosing our grid resolutions. We first defined a necessary resolution in the flow direction and transverse to the flow to capture the channel geometry. In case of channel flows, we then considered the surface velocity gradients at characteristic front flow velocities in the flow direction and transverse to the flow direction. We kept the ratio between cell length and cell width smaller than the ratio between these longitudinal and transversal velocity gradients and smaller than ten. In case of flows on a runout plane, we kept the ratio below 2.5. The vertical grid resolution was then defined by the available computational resources in a way that results were obtained within reasonable time.

The mesh size used for the water content sensitivity experiments increased from 1 mm cell height at the bottom to 4 mm cell height at a distance 25 mm above the bed. This height of 25 mm corresponds to the maximal surface elevation reached at the position of the laser measurement situated one meter downslope of the gate. The cell width was constant 1 cm and the cell length was 2.3 cm.

The curved channel experiment was modeled with 39 cells in the radial direction and a radial grading from 1 mm cell height and 2 mm cell width at the bed to 3 mm cell height and 2 mm cell width 6.5 cm above the bed. In the flow direction, the resolution was constant with a cell length of 5 mm.

The modeled USGS flume with a smooth channel bed consisted of approximately 4 million cells to model the channel flow. This led to a constant cell length of 28 cm, a cell width of 3.3 cm and a graded cell height from 0.7 cm at the bottom to about 1 cm cell height at 19 cm above the bed, which is the highest point reached by the free surface at the laser 32 m downslope of the gate. The runout was modeled with 10 million cells with the same vertical resolution and 5 cm cell sizes in the x and y directions.

The USGS flume with bumps was represented with 6.5 million cells on a refined mesh, resulting in 1.5 cm cell length, 1.25 cm cell width and 1.4 cm cell height at the bottom. Three cell layers above the bottom the mesh coarsened in lateral and

bed-normal direction to 4.4 cm cell length and 2.1 cm cell height. At a height of 32 cm normal to the bed, the mesh coarsened again in the horizontal direction and was continuously graded vertically; however, the corresponding cells were in the air phase of the flow except for the release body. During release, the upper part of the material lies within the coarse mesh, but during column collapse as the flow accelerates, this material transits into the finer mesh closer to the bed, where it starts shearing.

We performed a grid resolution sensitivity analysis with the modeled experiments of the water content sensitivity study, as described below.

## 3.3    Experimental validation of water content sensitivity

In our modeling approach, the rheology of the slurry phase (the fine material suspension) depends on its yield stress, which is known to be exponentially dependent on water content (e.g., Hampton (1975), O'Brian and Julien (1988), Yu et al. (2013),

Hürlimann et al. (2015)), with increasing exponents for higher clay fractions. Therefore, a minor variation in water content may cause a strong change in flow depths and run-out distance. Three experiments from Hürlimann et al. (2015) with high water-content sensitivity were selected. These debris-flow experiments were carried out by releasing 0.01 m³ of debris flow material from a 0.4 m wide reservoir into a 4.4 m long and 2 m wide, 30° inclined plane followed by a 2.5 m long, 2 m wide and 10° inclined run-out section (Fig. 1). The flume was covered by a rubber layer with a burling consisting of flat circular

discs of 4 mm diameter and about 0.3 mm height every 5 mm to increase roughness. The experimental sediment mixtures used for model validation only differed in water content (27.0%, 28.5%, and 30.0% by weight) and contained about 1.6% smectite, 8.8% other clay minerals, 27.8% silt, 47.7% sand and 14% gravel. The corresponding wet bulk densities were 1822, 1802 and 1722 kg/m³.

     All selected experiments were simulated using the same value of $\delta = 36°$ for the angle of repose of the gravel mixture. This

angle of repose was determined in a simple adaptation of the method of Deganutti et al. (2011) by tilting a large box with loose material until a second failure of the material body occurred. The model parameter $\tau_{00}$ was calibrated to fit the observed run-out length of the 28.5% water content experiment, and the two tests with 1.5% higher or lower water content were used to validate the sensitivity of the model to water content.

     The model adapts to a new water content by calculating a new Herschel-Bulkley yield stress (see von Boetticher et al.

(2016)). However, the free model parameter $\tau_{00}$ should be adapted to the new water content as well. An initial and simple approach chosen here is to first evaluate the relative effect that a changed water content has on the modeled Herschel-Bulkley yield stress $\tau_y$. In a second step, we apply the same relative change to the calibration parameter $\tau_{00}$. Let $\tau_{y-cal}^{w-new}$ be the Herschel-Bulkley yield stress calculated by the model for a new water content, based on the original value of the calibration parameter $\tau_{00-cal}$ that is not yet adjusted to the new water content. We reduced or increased the free model parameter $\tau_{00}$

according to

$$\tau_{00} = \tau_{00-cal} \frac{\tau_{y-cal}^{w-new}}{\tau_{y-cal}}, \tag{1}$$

where $\tau_{y-cal}$ denotes the Herschel-Bulkley yield stress as calculated by the model in the calibration test before the water content changed. This way, the change of the yield stress initially calculated by the model is also applied to the free model parameter $\tau_{00}$.

Based on the calibrated value of $\tau_{00} = 41.3$ Pa for an experiment with 28.5% water content (the calibration test), the rheologies of the two mixtures with 1.5% higher or lower water content were calculated using equation 1. For a water content of 27.0% (subsequently denoted as the reduced water experiment) this procedure resulted in $\tau_{00} = 51.8$ Pa, whereas for the 30% water content (denoted as the increased water experiment in the following) the result was $\tau_{00} = 33.5$ Pa. For each of the three experiments, laser-measured flow depths were available in the center of the flume, one meter downslope of the gate. Comparisons between measured and simulated flow depths at such small scales are only approximate due to the surface disturbance by coarser grains that cause significant fluctuations in surface elevation. However, the arrival time, the maximal flow depths and the decrease of surface elevation over time were considered to be suitable for comparison to the model. The model performance was evaluated by comparing the deposition patterns, travel times, and time series of flow depths in the simulations and experiments. The simulated flow depths reproduce the laser signal with respect to both time and amplitude (Figure 3) and predicted run-out deposits replicate the water content sensitivity (Figure 4) but underestimate the effect. Although the front arrival in the reduced water content test is delayed by 0.2 s in the simulation, the maximum flow depth is reached at the same time in the experiment and simulation (Figure 3 top). The maximum flow depth in this test is accurately predicted by the model, with a deviation of 2 mm, which is less than the average gravel grain size. The fast decrease of the measured surface elevation within 0.1 s after the peak is well captured by the model, followed by a moderate decrease until 1.2 s (Figure 3 top). At this point, the flow depth approached a level where it fluctuates around 11 mm, which corresponds to the maximum grain size. This transition begins later in the simulation and declines further to a modeled final deposit of 6 mm thickness; however, large measured fluctuations of flow depth are likely due to the coarser grains that the model does not account for. The predicted deposit length of 2.41 m in the simulation overestimates the experimental value of two meters (Figure 4 top).

There is an almost perfect fit between the shapes of the experimental and simulated deposits in the calibration case (Figure 4 center). The maximum flow depth of the calibration test and the subsequent flow depth decrease are well reproduced (Figure 3 center), although the front arrival time at the laser is again delayed by some 15 %. The measured and simulated flow depths in the experiment with increased water content show that the earlier front arrival time with higher water content is captured by the model (Figure 3 bottom), but the maximum flow depth is underestimated. The moderate decrease of the surface elevation over time is captured by the model (Figure 3 bottom). The final deposit thickness of about 4 mm at the laser is reproduced correctly but the run-out length of 4.08 m is under-predicted by about 15% compared to the experimental value of 4.84 m (Figure 4 bottom).

### 3.3.1   Grid sensitivity analysis

We simulated the three different water content experiments using a coarser grid resolution with twice the cell length, width and height, and conducted the same simulations on a finer mesh that reduced the cell width, length and height by one third. The reduced numerical costs of the coarser mesh allowed running the simulations once without recalibration and once with a recal-

**Table 1.** Comparison of modeled and measured runout distance $L$ for different mesh resolutions and water contents

| water content | L coarse mesh | L coarse mesh recalibrated | L fine mesh | L measured |
|---|---|---|---|---|
| 27% | 2.47 m | 2.36 m | 2.25 m | 2.00 m |
| 28.5% | 3.41 m | 3.23 m | 3.05 m | 3.20 m |
| 30% | 4.61 m | 4.31 m | 4.32 m | 4.83 m |
| water content | deviation abs., rel. | | | |
| 27% | 0.47 m, 47% | 0.36 m, 18% | 0.25 m, 13% | |
| 28.5% | 0.21 m, 7% | 0.03 m, 1% | -0.15 m, -5% | |
| 30% | -0.22 m, 5% | -0.52 m, 11% | -0.51 m, 10% | |

ibration of the experiment with 28.5% water content to the new coarse mesh, followed by adjusted coarse-mesh simulations of the 27% and 30% water content experiments using equation 1. We did not perform a recalibration with the refined mesh due to numerical costs. We only simulated the experiments on the finer mesh applying the original calibration parameter. Table 1 lists a comparison of the resulting runout distances.

Recalibrating the 28.5% water content experiment to the coarse mesh, we changed $\tau_{00}$ from 41.3 Pa to 45.0 Pa to achieve 1% precision in runout prediction. We repeated the adaptation of $\tau_{00}$ to the water content of 27% and 30% using equation 1, which led to a change of $\tau_{00}$ from 33.5 Pa to 36.5 Pa for the 30% water content mixture and to $\tau_{00} = 56.3$ Pa instead of 51.8 Pa in case of the 27% water content mixture. We address other aspects of the grid sensitivity in the discussion section.

### 3.4    Non-Newtonian rheology in channel bends: Evaluation of surface super-elevation due to curvature

Enhanced super-elevation due to curvature is characteristic for viscous debris flows (Wang et al., 2005; Bertolo and Wieczorek, 2005), so it can be viewed as a further indicator for model quality. Because we expect the enhanced super-elevation of debris flows in curved channels to be connected to a change of viscosity due to the pressure increase caused by deflection within the curve, the second group of experiments focuses on the pressure-dependent rheology. Enhanced super-elevation of debris flows were first modeled by Pudasaini et al. (2005) where the pressure (as normal load) increased as explicit functions of

slope curvature and twist. Such a model has further been extended in Fischer et al. (2012) where this aspect has further been explored by implicitly connecting the surface-geometry-induced curvature, and possibly also twist, to viscosity via its pressure-dependence (Domnik et al., 2013).

With $\beta$ as the average surface inclination transverse to the flow direction, a correction factor $k^*$ can be defined as the ratio between the gradient $tan(\beta)$ of super-elevation of debris flow material and the corresponding gradient of clear water with the

same average flow velocity. Based on the forced vortex approach with the assumption of a constant radius of the channel bend,

Scheidl et al. (2015) investigated the effect of flow velocity and super-elevation for several debris flow mixtures. However, for a mathematical derivation of the slope induced super-elevation we refer to Pudasaini et al. (2005).

The experiments were performed by releasing 0.0067 m$^3$ of material from a reservoir, through a transitional "box-to-channel" reach, into a channel of half-circular cross-section with 0.17 m diameter and a constant downslope channel inclination of 20° (Scheidl et al., 2015). The channel was arranged in an S shape, with a first 60° curve to the left with 1.5 m curve radius followed by a second curve to the right with 1 m curve radius (Fig. 5). The channel bed was covered with sandpaper to increase roughness.

Here, we consider the mixture with the largest clay content (mixture A of Scheidl et al. (2015)) where less demixing and phase separation was observed, and focus on the first curve of 1.5 m radius. The flow height in a cross section, two-thirds of the way through the curve, was measured by three lasers across the channel (Fig. 6). The arrival time at the laser section could not be used as a criterion for model calibration, because a simplification of the box-to-channel reach was necessary. This was due to the fact that the complex geometry at the transition to the channel (Fig. 5 region (d)) caused local air inclusions. A very fine grid resolution would be necessary here to adequately simulate the immediate demixing of the air. Therefore, as a simplification in our model set-up, the straight channel section was extended to the reservoir where it was filled with material at rest. Thus, the measured travel times between the gate and the first lasers are not comparable to those in the model. Instead, the free model parameter $\tau_{00}$ was calibrated to correctly predict the front velocity at the laser section. This front velocity was determined from high-speed video recordings. The average value from all experiments with mixture A at the upper curve was 1.49 m/s, leading to $\tau_{00} = 26$ Pa to reach the same flow front velocity in the simulation. Mixture A was composed of 6.5% clay, 15% silt, 26.1% sand and 52.4% gravel by dry weight. Since the gravel phase in the experiment was created from the same gravel as used in section 3.3, a value of $\delta = 36°$ was applied here, too. The water content was 27% and the density of the mixture was about 1800 kg/m$^3$.

The measured and simulated surface deflections can be compared to assess how well the modeled rheology accounts for the increased super-elevation. Nevertheless one should be aware that in this experimental setup, a granular front developed, which is in contradiction to the homogeneous phase distribution in the current implementation (Fig. 6 right). To account for the granular flow front, a mechanical phase-separation model like that described by Pudasaini and Fischer (2016) would be required, or even a coupled Lagrangian particle simulation.

The laser at the inside of the curve did not always register any material in the experiments or simulation, so the gradient of the super-elevation angle, $tan(\beta)$, is reconstructed from the simulation first as $tan(\beta_{max})$ by using the points of maximal surface elevations at the inside and outside of the flow ($P_i$ and $P_o$ in Fig. 6), and again as $tan(\beta_{min})$ based on the elevations of laser 2 and laser 3. At the moment of maximum surface elevation, the modeled $tan(\beta_{max})$ equals 0.336, resulting in $k^* = 2.11$ as defined by Scheidl et al. (2015), which fits the experimental average of $tan(\beta) = 0.33 \pm 0.05$ (Scheidl et al., 2015, table 2) and the corresponding correction factor $k^* = 2.1 \pm 0.6$. The corresponding value for $tan(\beta_{min})$ reaches 0.243, underestimating the experimental values; however, the corresponding correction factor $k^*$ equals 1.52 and still lies within the experimental standard error. The surface super-elevation is captured by the model although the front volume is underestimated by more than 50%. The under-predicted volume is a consequence of both the simplified geometry of the release area and the continuous over-prediction

of material losses at the channel margins where material becomes immobile due to the no-slip boundary condition, whereas in the experiment little material was deposited at the wall because the walls were moistened prior to the experiment. This problem needs to be addressed before more detailed comparisons can be made. Nevertheless this example is included to illustrate that the model can predict plausible degrees of superelevation.

## 3.5    Large scale experiments: Effects of bed roughness and share of fine material

Because it is difficult to upscale from laboratory-scale tests to true debris flow events, large-scale debris flow experiments are essential for model validation. The USGS debris-flow flume consists of a 75 m long, 2 m wide and 31° inclined concrete channel, with a release reservoir having the same width and slope, and an approximately 7.5 m long distal reach where the bed inclination forms a smooth transition to a run-out plane with a 2.5° inclination and no lateral confinement (Iverson et al., 2010). Laser sensors measure the flow height 32 m and 66 m downslope from the release gate, and a third laser is located in the run-out plane. The flume is tilted to one side, and the maximum tilt reaches 2°. The model accounts for this with a 1° tilt over the whole flume length.

We selected three experimental setups, two to illustrate the model capabilities and one that has a material composition where the model is only applicable with restrictions. The model cannot account for grain collisions and therefore is limited to debris flow mixtures which contain significant amounts of muddy suspension with a high loam content. The two first setups focus on experiments with released material consisting of the so-called SGM mixture (Iverson et al., 2010), which is composed of 2.1% clay, 4.9% silt, 37% sand and 56% gravel (by dry weight). Following James and Bait (2003), we assume that kaolinite dominates the clay minerals, although smectite could be present. We have chosen an SGM-mixture experiment with a documented runout deposit, and with a smooth channel bed surface to reduce the influence of granular collisions. The second setup should illustrate the interplay with the pressure-sensitive representation of the gravel, so a rough channel bed was used.

An important element of our simulation is that we compare the model with the channel experiments based on the calibration with respect to the arrival time of the flow front. Additional simulations that illustrate the model performance for simulating the so-called SG-mixture (Iverson et al., 2010) and the corresponding flow on smooth channel beds were conducted without recalibration, based on the calibration for the smooth channel SGM-mixture experiment (Fig. 7). The SG-mixture has no clay and is composed of 1% silt, 33% sand and 66% gravel.

### 3.5.1    Smooth channel experiment with high content of loam

The selected experiment was documented by Major (1997) as a release of 9 $m^3$. The water content and density of this test were determined as 18.5% and 1761 $kg/m^3$, respectively, by assuming fully saturated material. The angle of repose was estimated to be 39.3° on the basis of tilt table tests of the SGM mixture (Iverson et al. (2010)). The experiments were documented by videos and by surveys of the runout deposits (Logan and Iverson, 2013). The model parameter $\tau_{00} = 90\,\mathrm{Pa}$ was calibrated such that the time between release and front arrival at the run-out plane matched the experiment. Both the run-out process and the final deposit therefore contributed to the model validation, because they were not considered for the calibration. The simulated spreading into the run-out plane evolved in good agreement with the experimental observations (Fig. 8). In the experiment,

several surges arrived at the run-out plane after the time sequence shown in Figure 8, widening the material deposit at the foot of the channel. By contrast, the model evolved in a single surge. The maximum deposit length on the run-out plane in the experiment was 15 m and the simulated front reached 14 m.

### 3.5.2  Rough channel experiment with high content of loam

The SGM mixture was applied in the rough channel experiments with 17.9% water content and 2010 $\mathrm{kg/m^3}$ density (based on Iverson et al. (2010) table 2). The angle of repose was estimated to be 39.6° on the basis of tilt table tests of the SGM mixture (Iverson et al., 2010, see column (SGM) in table 3).

For the rough channel experiments, round-nosed cones as bumps of 1.6 cm height were installed on the bed every 5 cm; these were introduced in the model as pyramids as a trade-off between resolving the bed rougness and limitations of grid resolution due to numerical costs. Flow depth and basal force measurements were available as averaged values over a set of experiments with identical releases and channel setups, where the SGM material mixture (Iverson et al., 2010) forms a 9.7 $\mathrm{m^3}$ release body of known geometry. Three SGM experiments published by Iverson et al. (2010) with different flow front velocities were selected for front position comparison. A test from the year 2000 (test 000928) represented an extreme case with a quite low front velocity in the beginning followed by a sudden speedup of the flow front after 6 seconds. The other two tests also showed a sudden acceleration of the flow front, but their release process was faster and the sudden change in front velocity appeared later and was less dominant (Fig. 9). To some extent the difference in front position over travel time seems to be due to a large second surge that originates from the reservoir (Fig. 10). Especially in test 000928, part of the material left the reservoir with a delay, but as the second surge arrived at the front, the material front velocity doubled from about 8 m/s to 16 m/s.

The rough channel experiments with the SGM mixture were modeled with $\tau_{00} = 82.8$ Pa, leading to good agreement of the modeled flow front with the flow depth measurements at 32 m (see Fig. 12 $a$). In addition to the average friction angle derived from tilt table tests, a second simulation with a friction angle corresponding to the lower limit of measured friction angles was carried out. In this way, while keeping the value of the lower friction angle within the range of realistic values, we intend to balance out the overestimated roughness of the bumps by their modeled representation as pyramids. Both simulation results, i.e. $\delta = 39.6°$ and $\delta = 36.6°$, are shown in the diagrams to demonstrate that the effect is relatively small.

Flow front position, shape and surface wave patterns (derived from video recording) were compared to the corresponding simulations (Fig. 9 and Fig. 11), indicating a good agreement in front position and a comparable pattern of the small surface waves. The simulation with reduced gravel friction angle showed better agreement in the decelerating part of the flume, as expected. In the upper part of the flume, the modeled front seems to proceed too fast, which is due to the neglected gate opening process. However, a comparison of the modeled flow depths with the ensemble-averaged laser signal from all eight published SGM experiments 32 m downslope from the release gate shows that the simulated front arrived at almost the same time as the measured front (Fig. 12 $a$). Further downstream, at 66 m downslope, there is a discrepancy in measured and simulated front arrival times (Fig. 12 $b$), but the corresponding measured and simulated basal pressures fit well, especially for the simulation with a reduced gravel friction angle (Fig. 12 $d$). The flow depths generally developed within the standard deviation range of the measured values at 32 m, except at the late tail of the flow after 8 s from release, where both simulations resulted in some

overestimation corresponding to two slow surface waves passing. The simulation with the smaller friction angle of $\delta = 36.6°$ reduced the oscillation of flow depths compared to the simulation with the larger gravel friction angle. The measurements of three force plates installed at 31.7, 32.3, and 32.9 m were averaged over all eight SGM experiments and compared with the pressure in the corresponding cells of the simulation (Fig. 12 $c$), and the same was done for the basal force measurements at positions 65.6, 66.2, and 66.8 m (Fig.12 $d$). The model initially overestimates the pressure fluctuation, probably due to the simplified representation of the bumps at the bed as pyramids. However, after 7 s of flow the modeled basal pressures lie within the standard deviation of the measured values.

## 4    Discussion

This study represents an attempt to develop a widely applicable modeling framework for debris-flow simulations, based on rather simple constitutive equations describing the two-phase bulk flow rheology and combined with traditional 3D CFD modeling. Nevertheless the results are surprising, as it appears to be possible to accurately simulate front velocities, flow depths and run-out distances for the different material compositions and experimental setups by calibrating only one of the two free model parameters.

### 4.1    Modeled water content sensitivity

The simulations of the small-scale experiments that focus on water content sensitivity could reproduce the pronounced dependency of the run-out length on water content with some underestimation of the effect. The model could predict flow depth developments over time. Some short-time peak deviations between observations and simulations reached values close to the maximum grain size, possibly resulting from single-grain effects. The underestimation of the influence of the higher water content led to a run-out under-prediction by 15% in the model compared to the observation. The deposit of the calibration test case was accurately reproduced by the model, but the run-out of the reduced water content experiment was over-predicted by 17%.

### 4.2    Grid sensitivity

The discrepancy in runout length of the water content sensitivity tests could not be reduced with better grid resolutions for all the three water contents because the model showed a general trend to decrease the runout distance with increasing grid resolution. The mesh resolution study showed a consistent decrease in runout distance with increasing grid resolution. The modeled experiment with 28.5% water content on the finer mesh underestimated the runout distance by 15 cm or 5% whereas the coarse mesh without recalibration increased the runout prediction by 21 cm or 7%. The relative decrease in maximal runout due to the increased grid resolution, defined as (runout coarse mesh – fine mesh)/(average between runout coarse mesh and fine mesh), was 9% for the lower water content mixture, 11% in case of the calibration experiment and 7% for the increased water content simulations. The enhanced underestimation of the runout with 30% water content due to a fine grid resolution counterbalanced the slight improvement obtained on a finer grid in the reduced water content experiment.

In the reduced water content experiment, the mobilization of the release body was slower than in the experiments with higher water contents. In the 27% water content experiment, the front arrival time at the laser decreased with increasing grid resolution from about 0.6 s after release for the coarse mesh, to 0.8 s in case of the original mesh, and 1.2 s in the fine grid simulation. For this experiment, we integrated the modeled downslope velocity over the material volume close to the moment of front arrival at the laser. By dividing the volume-integrated velocity by the debris volume at this time step, we obtained a volume-averaged downstream velocity of about 1.3 m/s in case of the coarse grid and 1.1 m/s for the fine grid at the moment of front arrival at the laser. The corresponding volume-averaged slurry and gravel viscosities were 4.8 Pa·s and 7.2 Pa·s for the coarse mesh and 13.9 Pa·s and 9.3 Pa·s in case of the fine mesh. The pronounced difference between the two mesh resolutions, especially with respect to the volume-averaged Herschel-Bulkley viscosity, indicate higher shear rates on coarser meshes during release, which lead to faster flows due to the non-linear rheology. The recalibrated coarse mesh simulations indicate that the free model parameter can counteract the consequences of changing shear rates that are caused by altered mesh resolutions.

On the fine mesh, front fingering occurred before the material came to rest, which only appeared when using a Coulomb-viscoplastic rheology together with the volume-of-fluid method. The volume-of-fluid method, in general, tends to split the material into droplets when the flow depth becomes small. This effect remains even in hybrid approaches like the coupled level set-VoF method (Wang et al., 2008). The debrisInterMixing solver thus tends to develop splashes that separate from the main material body in case of shallow runout deposits. A multiphase model that solves one Navier-Stokes equation for each phase or a coupled Lagrangian particles simulation are needed to treat the development of the granular flow front accurately, but this would severely increase the computational costs.

### 4.3 Enhanced super-elevation in channel bends

The results of the curved channel experiments are encouraging, but further research is needed to evaluate whether the super-elevation of the surface due to channel curvature can be represented with such accuracy for other mixtures and at the lower channel bend. The focus on a mixture of high clay content was due to the fact that our simplified solver cannot account for phase separations due to grain-size sorting. The upper curve of the channel was simulated to save computational time. However, the non-Newtonian behavior resulting in increased super-elevation was more pronounced for mixtures with less clay content and slower front velocities at the lower curve. From the simulation results for the upper curve, we may at least conclude that the model can reproduce enhanced super-elevations and seems to be suitable for predicting debris-flow breakouts in curved channels for hazard assessment. Although only one set of comparisons with an experiment is shown, given the complexity of the problem and its intrinsic consequences, such results are important to highlight the potential applicability of the simulation model to geometrically more complex flows. The main limitation is that the maximum cross-sectional area of the simulated flow reaches only about 40% of the area determined from the experiments. We severely underestimate the debris flow volume at the curve due to the simplified release geometry and due to material sticking to the walls in the model due to the no-slip boundary condition, whereas in reality this material stayed mobile due to the wetting of the walls before the release. One could reduce the amount of material sticking to the walls by applying partial slip conditions like in Domnik et al. (2013), however, this would demand a multiphase approach to account for the wetting of the walls, which goes beyond the scope of this model.

Therefore, an improved mesh including the reservoir and the box-to-channel reach is necessary before addressing the flow in the lower curve, which is beyond the scope of this paper.

## 4.4 Simulation of large scale experiments with high and low content of fine material on a smooth and a rough channel bed

The large-scale experiments at the USGS flume were chosen as examples of flows closest to prototype conditions of the real world, with relatively small uncertainties concerning material composition, flow front velocity or run-out patterns. The experimental flow behavior was well captured by the model. In particular, the model successfully adapted from a mixture of 2.1% clay, 4.9% silt, with 37% sand and 56% gravel without recalibration to a material mixture without clay, containing 1% silt, 33% sand and 66% gravel, combined with a severe change of channel roughness.

For the smooth channel bed, the spreading into the run-out plane was examined in detail. In both the experiment and the simulation, the front arrived at the experimental maximum deposit length in comparable time, for both the so-called SG mixture and the SGM mixture (see Fig. 7 and 8). Although this indicates that the model captures the deceleration process with some precision, the lack of grain size sorting in the model clearly becomes apparent and viscous tail surges are not covered. This discrepancy possibly could be reduced with a phase-separation model. However, the modeled material deposition thickness in

the SGM experiment with a smooth channel bed is comparable with the experimental deposit in the front regions that were only covered by the first two surges (Fig. 8).

Three experiments of identical setup using the SGM mixture together with a rough channel bed were selected to compare flow front velocities with the simulation. Ensemble-averaged time evolutions of flow depths and basal pressures of eight such experiments were compared to the model output. The simulated flow depths lie in general within the range of standard

deviations of the measurements. However, considering the basal pressures, part of the deviations between experiment and simulation may arise from the pyramid representation of the bumps (round-nosed cones) in the rough channel bed, leading in the model results to overestimated pressure peaks and thereby to an exaggerated viscosity by the pressure-dependent gravel rheology. A reduced friction angle therefore improved the modeled flow front velocity, although the effect is not visible in the basal pressure fluctuation. The measured and simulated values do not agree in terms of the mean arrival times implied

by the laser signal at position 66 m (Fig. 12 $b$), however, when we use the basal pressure signal as an indicator of the front arrival, the measured and simulated arrival times fit well in case of the lower gravel friction angle simulation (Fig. 12 $d$). As stated by Iverson et al. (2010), the laser data were less suitable for arrival time estimates compared to the force plates, because the granular flow front included single grains that bounced ahead and were captured by the laser before the arrival of the dense material mixture. Because grain-size sorting effects and the release in two surges are not accounted for, a single

surge flow forms in the simulation, in contrast to the real tests where two surges formed in most of the experiments considered. Therefore, when the modeled debris flow reaches the end of the channel, the front composition and volume is not an adequate representation of the experiment. Phase separation effects would need to be taken into account by implementing either drift-flux models, multiphase approaches with one Navier-Stokes equation per phase or coupled Lagrangian particles or coupled discrete element methods. However, the corresponding model extension would introduce new model parameters and higher numerical

costs. As a consequence of our reduced approach without grainsize sorting effects, we did not model the run-out patterns of the rough channel experiments, in contrast to the smooth channel experiment where less demixing occurred.

On the one hand, it might be possible to obtain better representations of all SGM experiments with the current model by varying $\tau_{00}$ or density, water content, or the friction angle within the range of the published standard deviations of the experimental setup. On the other hand, we preferred to illustrate the model reliability based on input derived from averaged measurements, to avoid unrealistic expectations that cannot be fulfilled in practice. In future versions of the model it would be desirable to include grain-size sorting effects.

## 4.5  Contribution of the Coulomb-viscoplastic gravel representation within the flow process

The approach of a bulk-averaged viscosity derived from a Herschel-Bulkley representation of the fine material suspension and a Coulomb-viscoplastic representation of the gravel is based on the main assumption that the interstitial fluid can damp the grain collisions up to a degree where the tangential friction between gravel grains dominates the dissipation of the gravel phase. As an example at the limit of applicability, we have chosen a USGS flume experiment that applied a sand-gravel water mixture, in which the collision forces in general cannot be neglected. However, the selected experiment was conducted on a smooth channel bed, such that grain collisions were less pronounced and the video documentation shows a relatively dense material front where the grains are embedded in a slurry within the front. The experiment from the 21st of April 1994 is documented in Major (1997) with a release volume of 9.2 m$^3$, a dry bulk density at release between 1630 and 1960 kg / m$^3$ and a maximal runout length of 16.7 m. For the flow process in the channel, ensemble-averaged data for 11 such smooth-channel SG mixture experiments is available (Iverson et al., 2010). The average wet bulk density at release is 2070 kg / m$^3$ and the water volume within the release body averages 3.17 m$^3$ for the smooth-channel SG experiments (Iverson et al., 2010). We simulate the experiment by representing the sand suspension with the Herschel-Bulkley rheology with 16% water content according to the average numbers for the SG smooth bed experiments (Iverson et al., 2010, table 2 and 3). The gravel is covered by the Coulomb-viscoplastic rheology. We used the same simulation grid as for the SGM smooth bed simulation and applied the same $\tau_{00}$ value. The resulting Herschel-Bulkley rheology for the sand suspension had a density-normalized yield stress of $\tau_y = 0.0526$ m$^2$/s$^2$ and a corresponding consistency factor of k = 0.0158 m$^2$/ s$^{2.34}$. The Herschel-Bulkley exponent was chosen as n = 0.34. The volumetric share of sand suspension in the mixture was 57.5% and the gravel covers 42.5%. From the integration of the Herschel-Bulkley viscosity over the material volume at the moment of front arrival at the laser at position 32 m, we obtained a volume-averaged slurry viscosity of 14 Pa·s that contributes with 57.5% to the overall viscosity. The corresponding volume-averaged gravel viscosity was 54 Pa·s, contributing 42.5% to the overall viscosity. The modeled volume-averaged flow process 3.6 seconds after release was thus clearly dominated by the Coulomb-viscoplastic rheology. In a second step, we removed approximately half of the simulated material at 3.6 s after release by excluding cells with pressures over 1500 Pa, which led to a volume-averaged slurry viscosity of 21 Pa·s and a Coulomb-viscoplastic average viscosity of 49 Pa·s. Thus, the modeled material mixture was dominated by the gravel rheology even within the 50% of material that moved under lower pressures than the rest of the material. An adequate simulation of the experiment as achieved here (Fig. 12 $e$, $f$)

is not possible only using a Herschel-Bulkley rheology with the parameters linked to the material as in von Boetticher et al. (2016).

## 4.6 Advantages and limitations of the model

Our approach allows the model parameters to be linked to material properties and the model accounts for effects of the local topography on the shear stresses within the material. It suggests that one should be able to develop a model that can be applied to a wide range of debris-flow simulations, wherever the necessary data on material and site conditions are available. However, a one- or two-parameter model, although simple and cheap, may not capture more complex debris flows where complexities arise from different material or mechanical parameters and dynamically and locally evolving flow quantities (including the solid or fluid fraction and phase velocities, etc.). The purpose was not to gain a perfect representation of the experiment, but to see how the chosen rheology represents the sensitivity to water content, channel roughness and curvature, and fraction of fine material. The new model reduces the complexities of debris flow modeling: debris flow models commonly depend on many free parameters or are simplified by either modeling the flow from a granular perspective neglecting the interstitial fluid, or as a viscous continuum without accounting for the granular component of the flow process. Two-phase approaches, on the other hand, involve high numerical costs. In the case of two-phase coupling by drag between grain and fluid, the uncertainty in the drag between granular and fluid phases necessitates parameters that are difficult to quantify in case of the non-Newtonian suspension and non-spherical gravel grains. As a consequence, no previous modeling approach has succeeded in predicting debris-flow behavior across such diverse experimental settings as those examined here, by modifying only a single parameter. However, numerical costs are still high for accurate results. The application is suitable for situations where the detailed flow structure is required. While the simulation of the smooth channel debris flow experiment at the USGS flume required seven hours per second of flow using 32 processors on the WSL Linux cluster HERA (consisting of Six Core AMD Opteron 2439 at 2.8 GHz), the rough channel experiment demanded ten hours per second of simulated flow and 44 processors due to the high grid resolution. However, these estimates are conservative because we did not have exclusive use of the computing power of the cluster during these tests.

## 5   Conclusions

The three-dimensional solver DebrisInterMixing-2.3 (von Boetticher et al., 2016) combines a Coulomb viscoplastic rheological model for the solid phase with a Herschel-Bulkley model for the fluid phase for debris flow modeling. Here we describe validation tests of the solver. Based on published experiments we show that it is possible to calibrate the model using measurable properties of the material and two parameters, the Herschel-Bulkley exponent and the yield stress $\tau_{00}$, which may require calibration. The Herschel-Bulkley exponent was held constant for all simulations.

We demonstrated the wide range of applicability of our new numerical debris-flow model. The model concept follows the strategy of shifting from requiring user expertise in debris flow model calibration towards requiring information about the modeled site. The presented simulations of a wide range of different experiments lead to the following conclusions:

1. The material mixture can be characterized based on clay mineral composition, proportions of clay, silt, sand and gravel, angle of repose of the gravel, and water content. For debris flows with a high content of fine material, a single free parameter allows calibration to adjust the model to the grid resolution.

2. The model can account for changes in the material mixture and water content without recalibration.

3. The model can account for the pressure and shear-rate dependent viscous stresses and thereby captures the sensitivity of the material behavior to channel geometry, including the enhanced surface super-elevation of debris flows in curved channels.

4. The sensitivity to surface roughness is captured by the model and it can be varied without recalibration.

The need to calibrate only one parameter in this study greatly simplified the model calibration process. For flows with large proportions of fine material, the Herschel-Bulkley exponent may be chosen constant as well, which can save significant time in the calibration process while still providing a highly detailed and reliable model. Although such a minimally parameterized model may fit the real-world data less well than a highly parameterized model (perhaps because the latter is over-fitted), the time saved in calibration can be used to explore a wider range of material composition and site properties. Because such changes in model setup are translated into consequences for the flow physics by the model, the ensemble of such simulations could be used to outline the consequences of changes at the site. For example, a change in topography by a construction, a change in expected water content by a drainage or a change in expected debris flow compositions by a new gravel deposit could be addressed with the model to visualize the corresponding changes in expected debris flows. Recalibrated models cannot deliver such information. Furthermore, in our model there is less room for the user to make arbitrary parameter settings than in models with several calibration parameters. Thus it may be possible to quantify the model's reliability in a robust and general way, because different users are likely to apply comparable parameter settings. However, one missing element is phase separation due to grain-size sorting effects, which would not only enable simulation of the granular front but also could enhance the model's capability to perform channel bed erosion by mobilizing gravel deposits. This extension may be included in future versions of the model.

## 6 Code availability

The source code can be downloaded from the supplement application.zip; please follow the instructions given in the README.pdf file for installation.

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

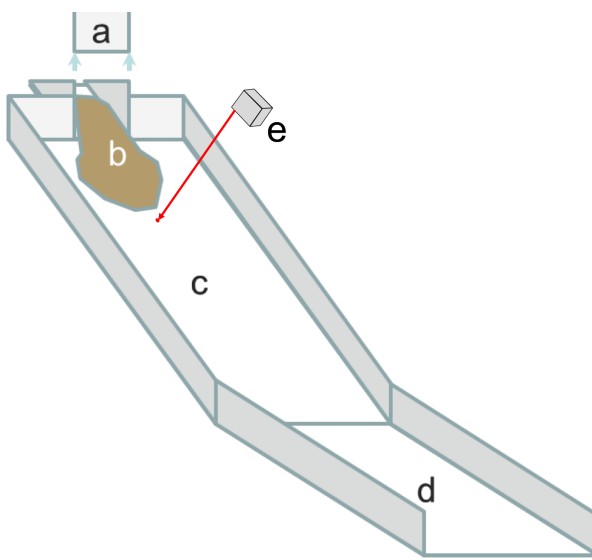

**Figure 1.** Isometric sketch of the hill slope debris-flow flume. Material (b) is released from the reservoir at the top by a sudden vertical removal of a gate (a) and flows down a steep slope (c) followed by a gently inclined run-out plane (d). The front arrival and flow depth are measured at the center one meter downslope of the gate with a laser (e).

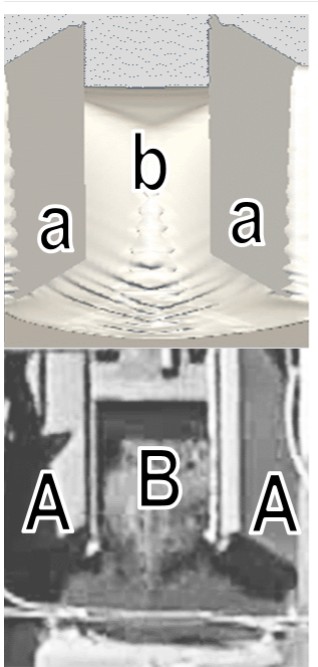 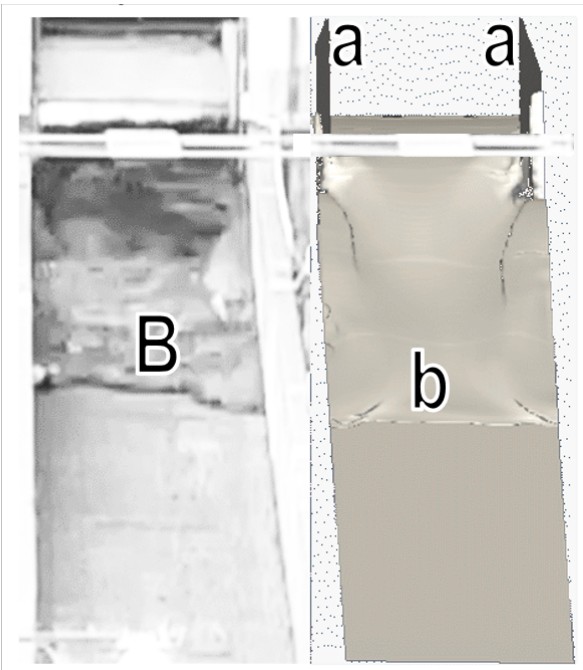

**Figure 2.** Screenshots comparing the modeled release, including the gate at the USGS flume, to camera pictures 0.7 s after release (left) and 1.7 s after release (right). The modeled gate wings (a) were introduced with a body force approach. They received accelerations over time which we derived from the gate wings' motion (A) in the video documentation (Logan and Iverson, 2013). Initially, a narrow centered flow front develops due to the opening (b and B on the left) which then widens laterally. Once the flow front reaches the sidewalls, it is reflected, which causes small surface waves that travel transversal to the flow from the sidewalls towards the center. These surface patterns occurred both in the simulation and in the experiment (b and B on the right, more apparent in the corresponding video from which the screenshots ar taken, see Logan and Iverson (2013)).

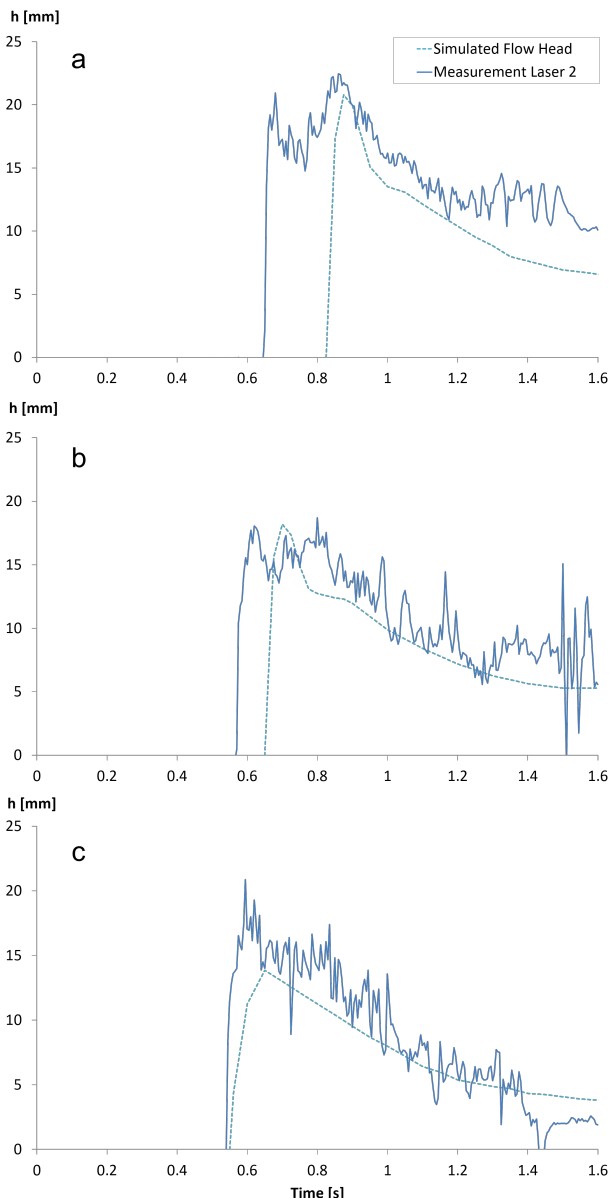

**Figure 3.** Laser measurement and corresponding simulated values of the flow depth over time, one meter down-slope of the gate for experiments with water contents of 27% (a), 28.5% (b) and 30% (c). The laser data were box-averaged over 10 milliseconds.

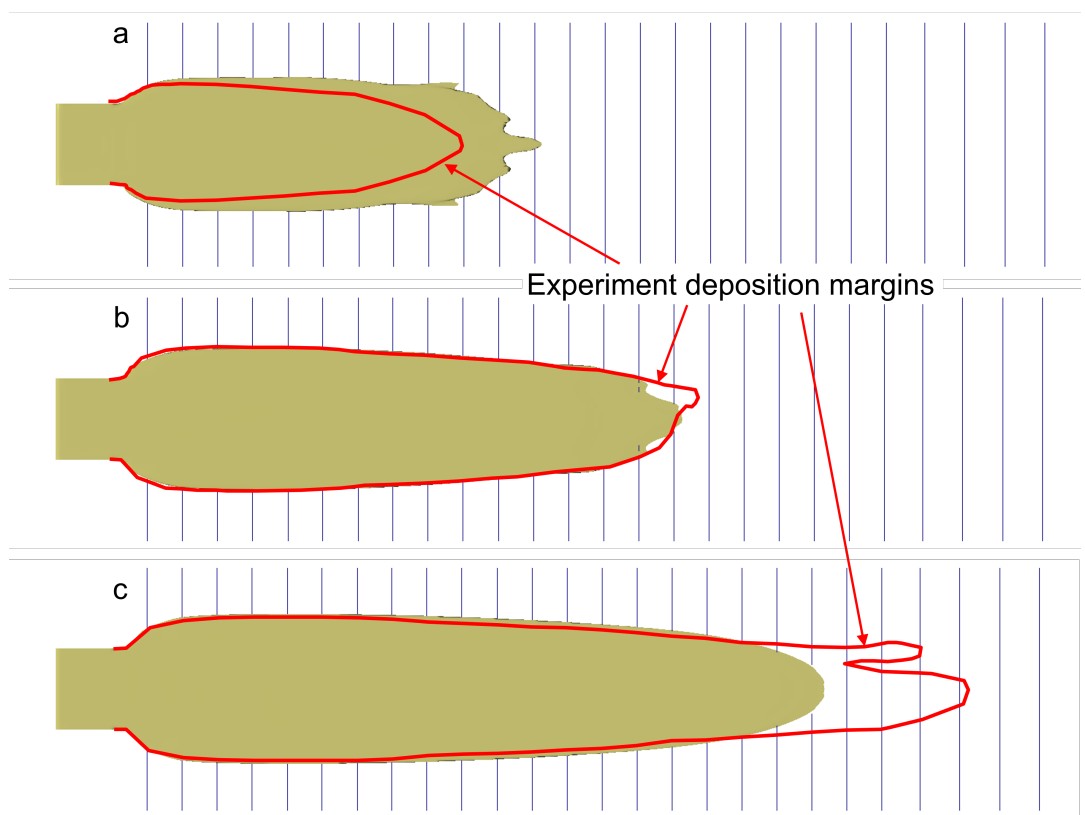

**Figure 4.** Simulated deposits (brown areas) and corresponding experiment deposition margins for mixtures with 27.0% (a), 28.5% (b) and 30% water content (c) applying $\delta = 36°$ and $\tau_{00}$ based on the calibration case of 28.5% water content. The vertical thin lines represent the 20 cm spacing marks in the hillslope debris flow flume (Fig.1).

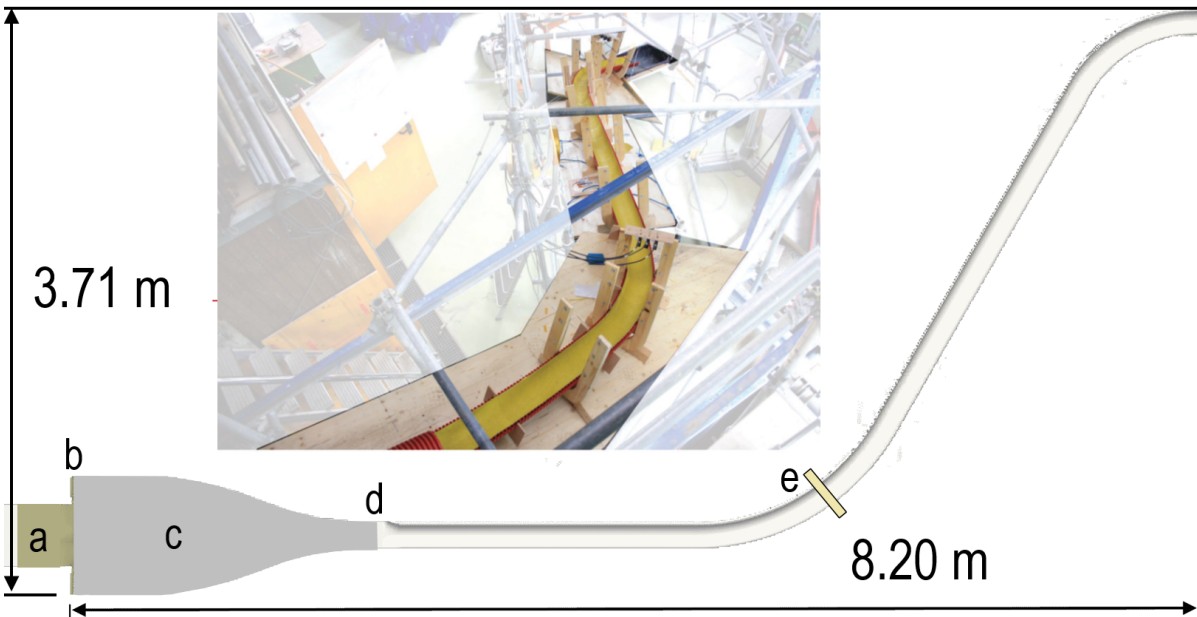

**Figure 5.** Photograph and a sketch with a top view of the modeled curved channel, with material released passing from a reservoir (a) through a flap gate (b) into the plane transition box (c). At the restriction (d) the channel profile changes from a rectangle to a half pipe. Three lasers positioned in a profile 40° after the beginning of the curve captured the surface elevation in the channel bend (e).

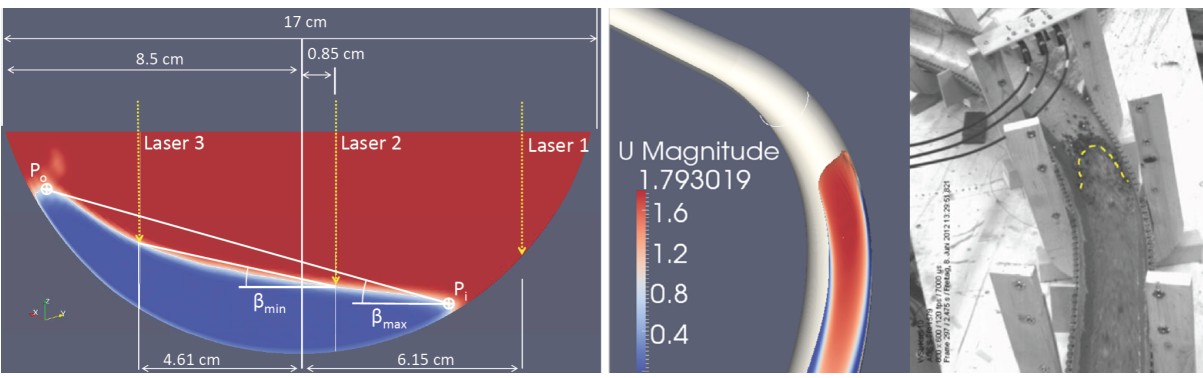

**Figure 6.** Left: View upstream on modeled channel section 40° after the beginning of the curve, showing air (red), debris mixture (blue), laser positions and minimal and maximal super-elevation angles $\beta$. Right: Screen-shot of modeled and experimental flow surface before reaching the laser section at the upper curve. The color bar denotes simulated surface velocity in m/s; the dashed line in the experiment indicates the transition between the granular front and the viscous mixture.

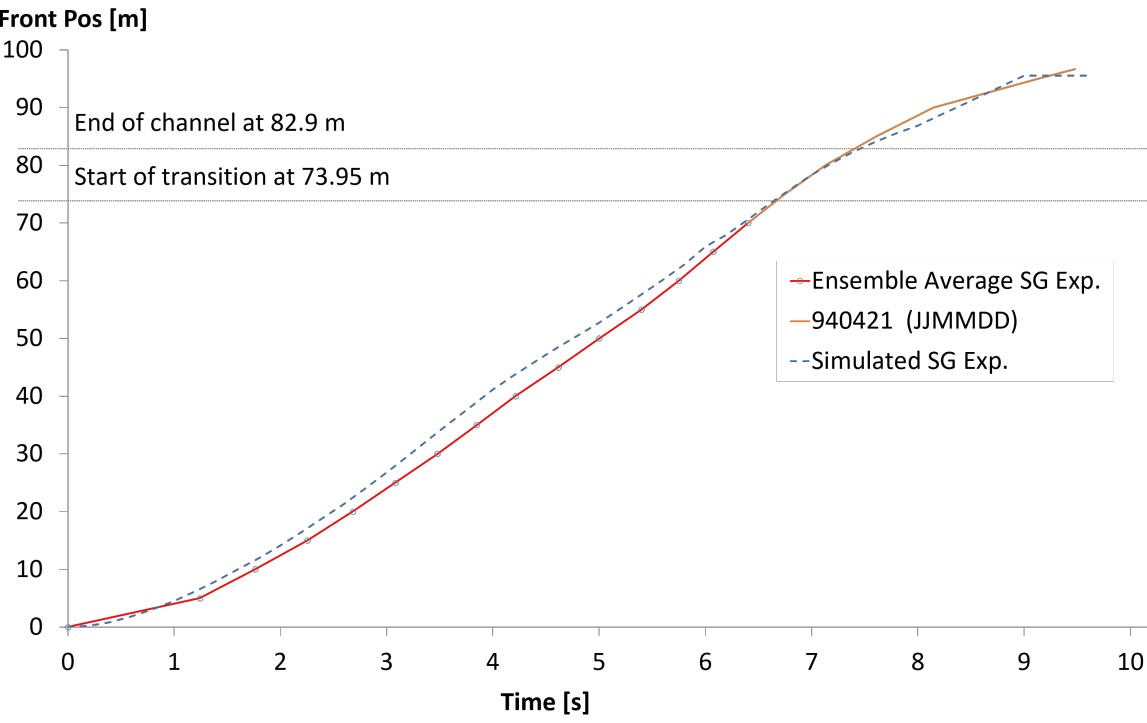

**Figure 7.** Comparison of the flow front position over time in the smooth-channel SG mixture simulation with the ensemble average front position up to 70 m flow distance as published in Iverson et al. (2010), and a continued comparison from 70 m on for the runout with an experiment from April 1994 (continued graph 940421). The value of $\tau_{00} = 90$ Pa derived for the SGM mixture was applied for the SG mixture without recalibration, using the Herschel-Bulkley rheology with 16% water, no clay, 1% silt and 33% sand.

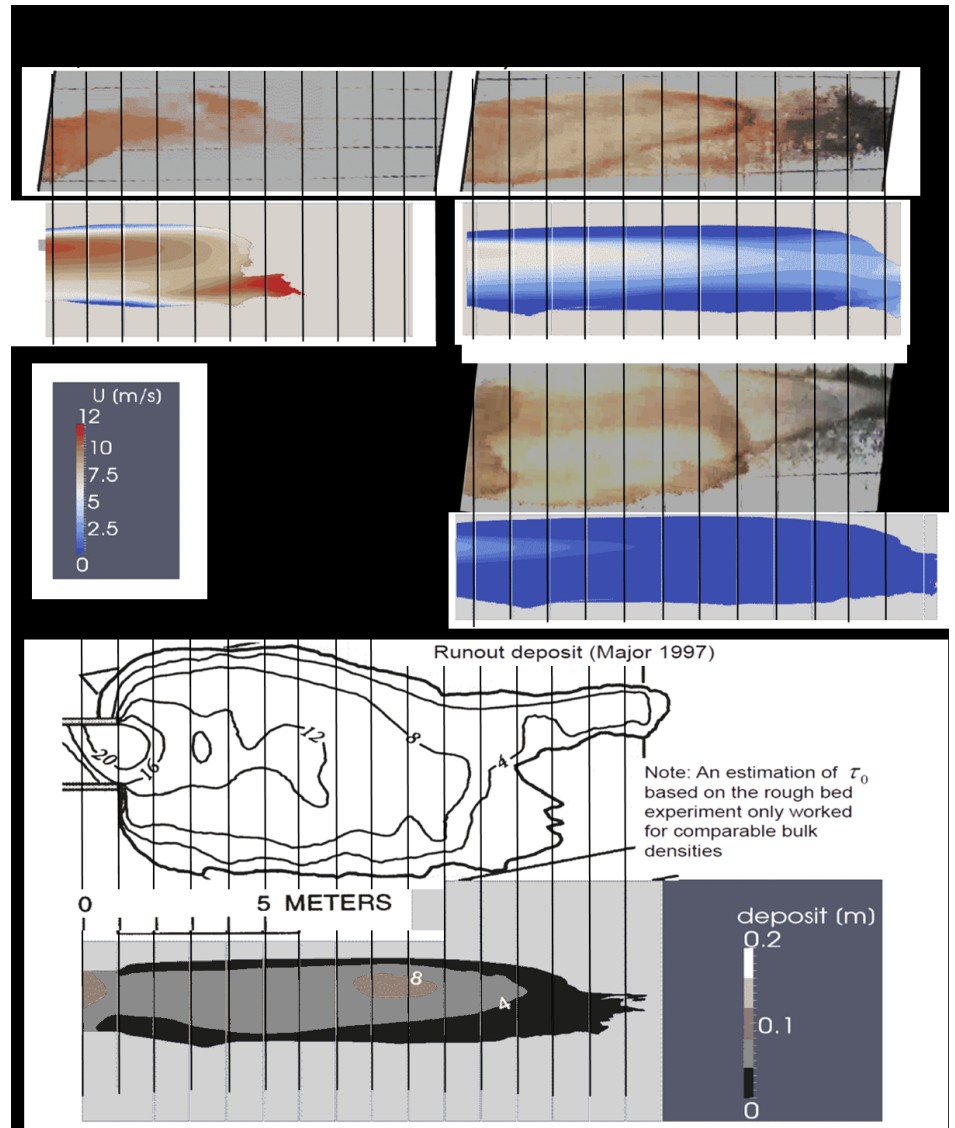

**Figure 8.** Time-tracking of the run-out process by top camera (top panel) and comparison of the final runout deposit in the smooth-channel SGM experiment (experiment 15 Major (1997) conducted at 05. 26. 1994) and simulation (bottom panel). The corresponding simulated material is colored by flow velocity (top) and deposition thickness (bottom). The free model parameter $\tau_{00}$ was calibrated to 90 Pa to fit the arrival time in the simulation to the experiment. The experimental time was derived by counting the number of video frames in the overview camera video between release and arrival in the run-out plane (Logan and Iverson, 2013). The deposit in the experiment widens up as later surges of liquid material meet the first surge deposit and spread to the sides at the beginning of the plane. The simulation only covers a single surge runout with homogeneous (unsorted) material and does not reproduce phase-separation.

**Figure 9.** The flow front position over time of the rough-channel SGM mixture simulations, compared to three selected replicate experiments using the so-called SGM mixture, a standard mixture of sand, gravel and loam (Iverson et al., 2010). We selected the three tests because the corresponding travel times could be derived from the published figure in Iverson et al. (2010). We investigated the development of the flow front in combination with the video documentation. The three tests experienced increasing front flow velocities after 6 seconds (black graph, test 000928), 7.4 seconds (pink graph, test 030625) and about 7.5 seconds (red, test 010913) as a second surge reached the flow front (observed from video, see Fig. 10). Since the bed roughness is overestimated by a representation as pyramids instead of round-nosed cones, one simulation used a reduced friction angle of 36.6° that corresponds to the lower boundary of possible experimental values based on published standard deviations (see Iverson et al., 2010, table 3).

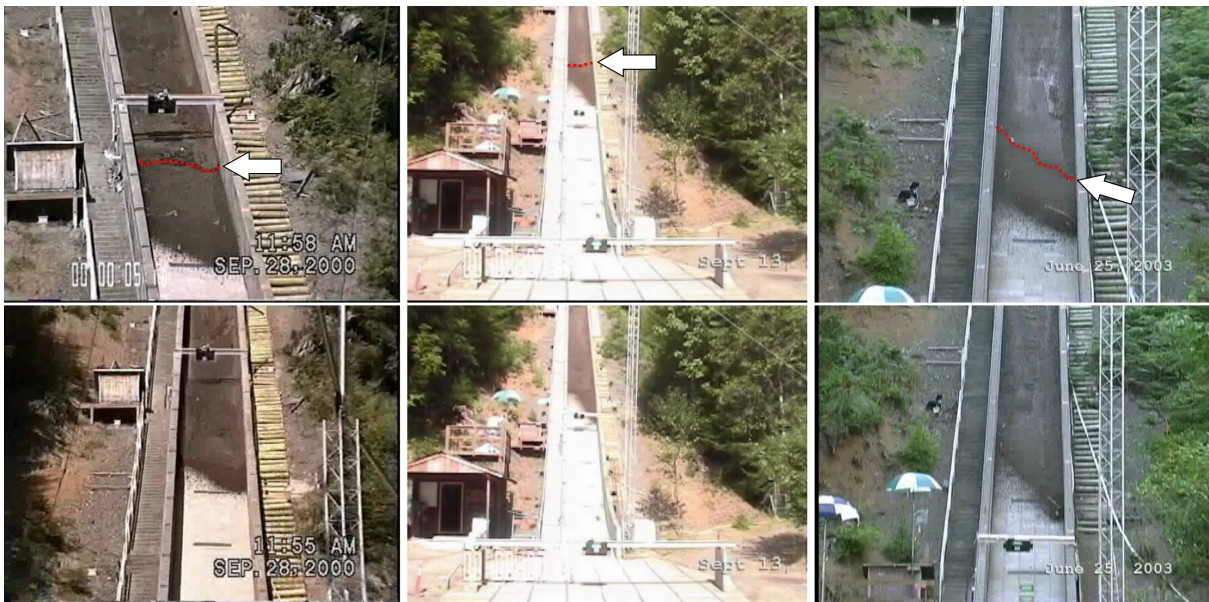

**Figure 10.** Snapshots of the flow front of test 000928 (left) 5.6 s after release (bottom left) when the approaching second surge (top left) unified with the front. The same happened in test 010913 (center top and center bottom) 7.4 seconds after release and in test 030625 (right top and right bottom) 7.0 seconds after release. The upper row of images capture the moment about half a second earlier in time than the lower row, showing the approaching second surge (indicated by the white arrows and a red line, the surge is more apparent in the video, see Logan and Iverson (2013)). In the lower row, the moments are shown were the second surge reaches the flow front. The corresponding time coincided with a sudden increase in front velocity; see Fig. 9.

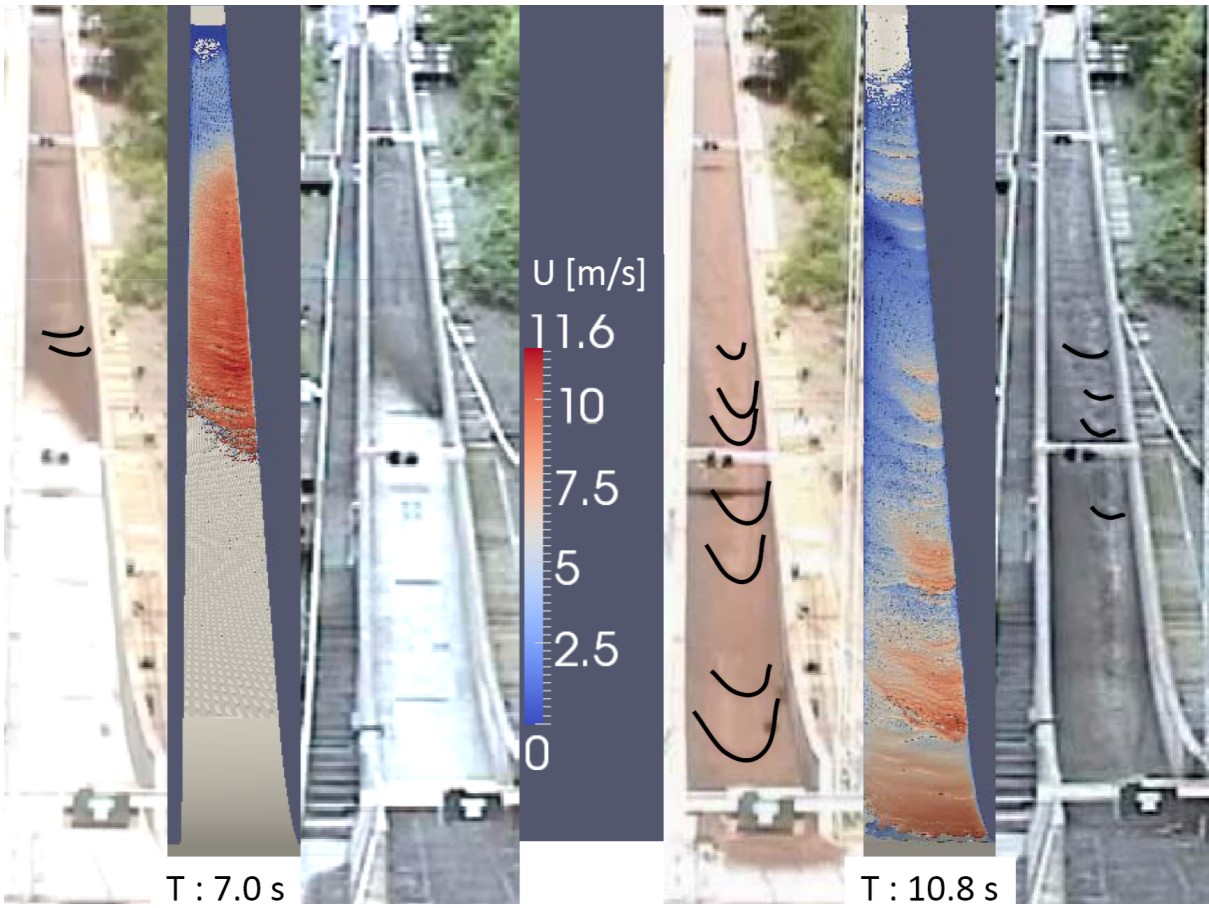

**Figure 11.** Comparison of the material surface in test 010913 (left of simulation) and test 030625 (right of simulation) at an intermediate position 7 s after release (left) and at the simulated front arrival at the run-out plane at 10.8 s (right) for a modeled friction angle of 36.6°. Black lines in the pictures highlight surface wave fronts, which appear more clearly in the video (Logan and Iverson, 2013). In the simulation such waves are indicated by encoding the surface velocity as the color scale.

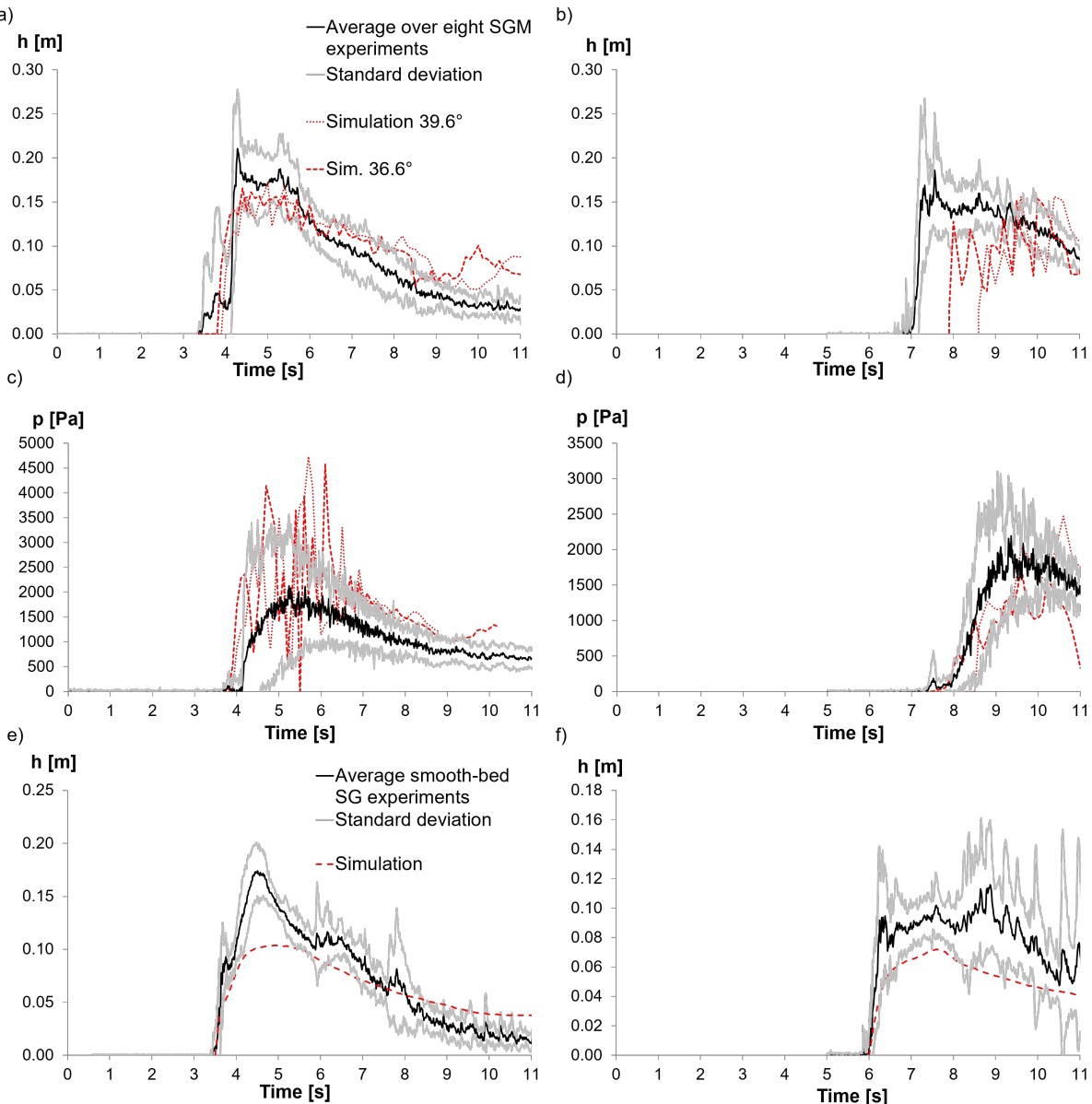

**Figure 12.** USGS flume experiments compared to the simulation with $\tau_{00} = 82.8$ Pa: (a) and (c) show the flow depth and basal pressures from the ensemble average of eight SGM experiments on the rough channel at a position 32 m downslope of the release gate, and the corresponding simulation results for two gravel friction angles. The diagrams (b) and (d) show the same for a position 66 m downslope, and (e) and (f) show the comparison of simulated and measured flow heights 32 m and 66 m downslope of the release gate for the ensemble average of the smooth-channel SG-experiments. The ensemble averages are based on the data published in Iverson et al. (2010). The measured basal normal stresses were derived as the average temporal value of three force plates placed in the channel center line at 31.7, 32.3 and 32.9 m downslope from the release gate (c) and 65.6, 66.2 and 66.8 m downslope (d). The force plates were circular; however, due to the simulation grid geometry, three squares of basal cells with the same areas and positions as the force plates were used to derive the values of basal pressure in the simulation.