# Peer review of "DebrisInterMixing-2.3: a finite volume solver for three-dimensional debris-flow simulations with two calibration parameters - Part 2: Model validation with experiments"

_Geoscientific Model Development, 2017_

## Referee Comment (RC1) · G. Chambon (Referee) · 21 Apr 2017

This paper discusses the capabilities of a new 3D finite-volume model developed for simulating the propagation of debris flows that are rich in fine particles. The model relies on a monophasic approach for describing the mechanical behavior of the material. The originality, however, is that the contributions of the interstitial slurry (water and fine particles) and of the large particles are handled independently, using distinct constitutive models, and then lumped into a global apparent viscosity through a concentration average. Owing to this approach, the authors claim that, once the model is calibrated

for a given composition of the solid material, it should be capable of accounting for changes in water content without further recalibration. The model is presented in detail in a companion paper recently published in the same journal. The present paper focuses on comparisons with experimental results of the literature at different scales.

Overall, the agreement with the experiments appears reasonably convincing, and shows the potential of the model. Since the code is provided as supplementary material (I did not actually test it), the readers will have the possibility to further explore the capabilities of the tool. The choice of the case studies appears a bit questionable, since none of them really challenges the 3D character of the model. A case of impact against an obstacle would have probably been better suited for that purpose. Yet, the presented case studies are interesting and allow the authors to evaluate the rheological module, which is the real focus of the paper. The comparisons with experiments are discussed in a balanced manner, highlighting the assets as well as the limits of the lumped approach. However, a more thorough discussion of the respective roles played by the slurry and granular contributions would have been interesting (see comments below). The paper is well-written and the presentation is already well-advanced, although the number of figures is probably in excess (see suggestions for removals below). Globally, I thus support the future publication of this paper in GMD, but I have first a number of specific and technical comments that I would like the authors to address in a revision of their manuscript.

Specific comments

1/ Nowhere do the authors properly specify how the distinction between particles belonging to the interstitial slurry (characterized by a Herschel-Bulkley rheology) and particles belonging to the granular phase (characterized by a pressure-dependent rheology), is made. I assume that this distinction is based on a grain-size threshold? What is then the value of this threshold? Typically, is the sand fraction considered to be part of the slurry or the granular phase? Since colloidal interactions are generally negligible for particles larger than a few microns, I would personally consider sand as belonging

to the granular phase. However, as the authors systematically refer to this granular phase as "gravel", one is led to think that sand is instead accounted for in the slurry. This issue would need to be better explained and discussed.

2/ Related to the previous comment, it would be interesting to fully explicit the computation of the lumped rheology in at least some of the examples treated: i.e., give values of the "full" yield stress of the slurry \tau_y (and not only of \tau_00); give values of the effective viscosities of the slurry and granular phases, and of the lumped material (concentration average), for representative shear rates and pressures. More generally, a discussion on the interest of considering this composite constitutive law in the examples shown would be interesting. Is the contribution of the granular phase significant? Would have it been possible to obtain equivalent results with only the viscoplastic part?

3/ I did not really understand the rationale behind equation (1) used to evolve the parameter \tau_00 with water content. In principle, one would expect this parameter to remain constant for a given composition of the solid material, and thus independent of water content. If I understood well, the authors do nevertheless consider a variation of \tau_00 with water content due to a sensitivity of their computations, in particular shear rate, to grid size. From my point of view, this issue should be discussed in more details (see also comment 4 below), and equation (1) should be better justified. In some sense, it can seem disappointing to develop a full 3D model relying on supposedly physically-based constitutive models and, in the end, to use such a trick to resolve what seems to be a purely numerical issue. In particular, the grid-size-sensitivity probably implies that the vertical structure of the flow is relatively poorly captured in the presented simulations. What is then the benefit of the 3D model compared to a depth-averaged approach?

4/ Nothing is said concerning the mesh characteristics used in the different examples presented: grid size, number of elements in the horizontal and vertical directions, etc. This is essential information for any reader interested in performing similar studies with the provided code. Furthermore, the influence of grid size on the presented compar-

isons with experimental results would also need to be discussed. I wonder, in particular, whether the results presented in Fig. 5 for the reduced and increased water contents (compared to the calibration case) could be improved with a finer mesh. Same question for the results presented in Fig. 15, notably the strong unphysical oscillations displayed by the pressure signal.

Technical comments

P. 2, l. 7: "Material properties are related to the fractions of different minerals...". Maybe specify "clay" minerals?

P.3, l. 25: "Therefore, for each material composition there should be a critical range where a minor variation in water content causes a strong change in flow depths and run-out distance." I do not really understand this statement. How is this "critical range" related to the exponential variation of the yield stress with water content?

P.5, l.1-9. The relative error (in %) between the simulated and experimental deposit length is indicated for the case of increased water content, but not for the case of decreased water content (in which case only absolute values are given).

P.6, l.11-12. Please also indicate the experimental values of tan(\beta_min) and of the corresponding correction factor.

P.7, l.5-7. Why were simulations of the SG mixture based on the same calibration parameters as for the SGM mixture? Since the composition of these two materials is different, it seems that a recalibration would be necessary? Furthermore, results obtained with SG mixture are not really described in the following (and Fig. 8 is never properly discussed). What is then the point of introducing this additional case?

P.10, l.21-23: "The measured and simulated values do not agree with the mean arrival times implied by the laser signal at position 66 m (Fig. 15 b), however, they do by means of basal pressures for the lower gravel friction angle simulation (Fig. 15 d)." Unclear sentence.

P.11, l.1. I do not fully understand what the authors mean by "Our approach allows the model parameters to be linked to (. . .) local topography".

P.12, l.1: "The model can account for the sensitivity of the rheology to channel geometry . . .". This is a strange statement: one does not expect the rheology (a material property) to be sensitive to channel geometry.

P.12, l.8-10: "Because such changes in model setup are translated into consequences for the flow physics by the model, the ensemble of such simulations may mirror how the modeled site would respond to similar changes." Unclear sentence.

Fig. 1: This figure may not be necessary.

Fig. 2: What are we supposed to see in this figure? What conclusion can be drawn from it concerning the influence of the gate on flow dynamics? To highlight the role of the gate, it would certainly be more demonstrative to show, e.g., front position versus time for cases with and without gate.

Fig. 7 top: This figure may not be necessary.

Fig. 10: This figure, on which details are hard to see anyway, may not be necessary.

Fig. 11. It appears that this Figure is never called, nor discussed, in the text. Is it then necessary? If the authors decide to keep it, axes and scale should be indicated. What is the thin vertical line that can be seen behind the thick red mark?

Fig. 13. This figure may not be necessary.

Fig. 15. It appears that plots e and f, corresponding to the SG mixture, are never called nor discussed in the text.

---

## Referee Comment (RC2) · Anonymous Referee #2 · 6 May 2017

The present manuscript applies the model and simulation technique described in Part I by von Boetticher et al. (2016) that is based on an adaptation of the interMixing-Foam, OpenFOAM solver in combination with a stable implementation of the pressure-dependent rheology model of Domnik et al. (2013) to describe the gravel phase as a Coulomb-viscoplastic fluid, combined with a Herschel-Bulkley rheology implementation for the interstitial slurry of water and fine sediment. Three different experimental setups are chosen to illustrate how sensitively the modeled flow and depositional processes react to changes in water- and clay-content in the mixture, channel curvature and roughness. The first experimental validation is based on flume experiments

from Hürlimann et al. (2015). The second experimental case used for validation was designed to study the sensitivity of debris flows to channel curvature (Scheidl et al., 2015). The third, with large scale out-door USGS flume experiments for debris mixtures. The audience of GMD may benefit from the publication of this MS. However, it needs a major revision.

As the present MS is the application of Part I, the Introduction/Discussion should briefly mention the need of the full 3D simulations, modeling assumptions, simplicity for application as well as the scopes/limitations of the modeling and simulation approaches as mentioned in Part I. This would help the audience who may only focus on application, to directly follow this paper.

Writing could be substantially improved in concept and content. Figures are not enough explained in the text. Figures quality and arrangements could be improved. Some important dynamical aspects observed in the simulations would have been explained in a better way with elaboration. Several figures/texts are not needed, so could be removed.

Some of the experiments considered here contain relatively low water content, however in reality debris flows may contain large amount of water, and may transform to debris floods, or locally fluidized. It would be appropriate to discuss this aspect that would probably highlight both the strength and limitations of the presented model and simulation aspects in relation to the experiments. In a debris flow body, water contain may evolve strongly (Pudasaini and Fischer, 2016; Mergili et al., 2017), and the characteristic may range from dense to dilute flows. These aspects need to be clearly mentioned in the MS. Recent and relevant literatures could be included and discussed.

Further detailed suggestions for possible improvements:

Title: Part 2: Model validation –> Part 2: Model validation with experiments

Abs.: material properties were known –> material properties and compositions were

known

(including ist mineral composition): R (R = Remove).

(including its friction angle): R.

two model parameters are sufficient for calibration –> two model parameters are used for calibration

the angle of repose: is this 'the angle of repose' of 'internal friction angle'?

Main text:

P1:

L14: a partly fluidized mass –> a partially or fully fluidized mass

The mix of –> The mixture of [?]

L17: Would be relevant to include recent works by Kattel et al. (2016, Anal. Glaciol.), Mergili et al (2017, GMD)

P2:

L3: Coulomb-viscoplastic rheology –> Coulomb-viscoplastic rheology (Domnik and Pudasaini, 2012; Domnik et al. 2013)

L7: (I= Insert): 'For the dynamic evolution of the solid and fluid concentrations and phase velocities we refer to more general model and simulations (Pudasaini, 2012; Mergili et al., 2017).'

L9: The object of this study –> The objective of this study

L10: sensitivity to water content, gravel- and clay-fraction –> sensitivity to water content, gravel- and clay-fraction (also see de Haas et al., 2015, JGR).

L11: the interaction between the three-phase rheology –> the interaction between the bulk rheology of the mixture

[Figure]

L15: Here more clearly and quickly mention about Part I

L23: The only parameter –> With this, the only parameter

L24: a restricted multiplication factor: Explain.

L28: 'side-wall effects': What is the difference in side-wall effects in single-phase flows and a kind of three-phase mixture flows of gravel, sand and water as considered here? Better to explain.

L33: I: We mention that super-elevation has been analytically modeled and validated for dry granular flows and flows of mixtures by Pudasaini et al. (2005, NHESS; 2008, PoF) with detailed discussions on the effect of channel curvature and twist for different channels and fluid fractions in the dynamics and super-elevation of the flow.

P3:

L1: sediment –> sediment water mixture

L4: flume –> two-dimensional rectangular flume

L5: a planner run-out area –> a planner three-dimensional run-out area

L9: is linked –> is often linked

L21: I: Since, in the mixture, largely the solid particles exhibit slip, viscous fluid exhibits no-slip along the basal surface such distinct basal boundary conditions can only be included with real two-phase mass flow models (see, e.g., Pudasaini, 2012; Mergili et al., 2017).

P4:

The mixture effectively consists of water, fine particles and gravel with different physical parameters and mechanical, hydrodynamical response to applied loads. In such a complex situation, how a simplified model with two free parameters can capture the flow so nicely. It needs to be discussed.

L1-2: 'angle of repose' and 'friction angle': they are not the same. One of them should be used consistently.

L6-11: Improve.

L11: According to equation 1, –> according to

L12: Bit strange notations.

L23: decay –> decrease [?]

L25: flow depths –> hydrographs

L26: (Figure 5). –> (Figure 5). Effect of the water content and geometry on the flow dynamics and super elevation have also been previously simulated by Pudasaini et al. (2005).

L31: a transition began towards the stable level of about 11 mm flow depth –> a transition began towards fluctuation around 11 mm flow depth [also explain panel (b) and (c) . . ..].

Even if the flow height is twice the maximum grain size what about the experimental simulation reliability/reproducibility/accuracy?

P5:

L1-2: However, the simulated front also temporarily paused at x = 2.04 m, until it was overrun by a second wave 0.1 s later: Not shown.

L2-3: There is an almost perfect fit between the shapes of the experimental and simulated deposits in the calibration case (Figure 5 center).: Explain the reliability of the perfectness, because the other two panels do not so strongly support this statement.

L3-4: The maximum flow depth and the subsequent decrease are well reproduced (Figure 4 center): There is no flow depth here; in Fig. 4 hydrograph, in Fig. 5 only deposition areas are shown.

L3-9: There is a non-logical switching between Fig. 4 and Fig. 5; difficult to follow.

L5: in the wet experiment: both experiments were wet, or?

L12: I: However, super elevation also occurs in dry granular flows as this phenomenon is primarily induced by the geometry of the channel (curvature and twist) rather than the viscous or frictional properties of the material. Nevertheless, super elevation is amplified by the fluid component in the mixture flows (Pudasaini et al., 2005, 2008).

so it can be viewed as a further indicator for model quality. –> so it can be viewed as a further indicator for model quality (from geometric point of view).

L14: pressure-dependent rheology. –> pressure-dependent rheology. This in fact has first been analytically and explicitly included in Pudasaini et al. (2005) where the pressure (normal load) increases as explicit functions of slope curvature and twist. Such model has further been extended in Fischer et al. (2012). This aspect has further been explored here by implicitly connecting the surface geometry induced curvature, and possibly also twist, to viscosity via its pressure dependence (Dominik et al., 2013).

L18: I: However, for the rigorous mathematical derivation of the slope induced super elevation we refer to Pudasaini et al. (2005).

P6:

L6: (Fig. 7 right). –> (Fig. 7 right). For this a mechanical phase-separation model (Pudasaini and Fischer, 2016) would be required.

L7: in the experiments or simulation –> in the experiments

L14: the front volume: not clear/not seen.

L21-22: Does it upscale?

P7:

L18: Separation between solid- and fluid-type materials may lead to this discrepancy

that can be described with phase-separation model (see literature mentioned above).

L23-32: Description could be improved.

P8:

L8: I: in the early stage of front arrival but substantial later.

L14-15: without any time delay. –> almost with no time delay.

L15-16: I: This discrepancy could have been emerged due to the fact that there could be substantial interactions and also separation between solid- and fluid-type phases that has not been considered in the simulations.

L27: a widely applicable –> a potentially widely applicable [needs to be applied to different flows and real events].

L28: two-phase flow –> two-phase bulk flow

L29: are surprising, as it appears to be possible to produce accurate front velocities –> produce appreciable front velocities

P9:

L7-8: Only grid-resolution (numerical) is explained as a possible source of discrepancy. But this discrepancy could also be reduced by applying real two-phase models with explicit phase-interactions. Needs discussion.

L20: As mentioned previously other relevant works could be discussed.

L21: Discuss the work by de Haas et al. (2015) and for grain sorting and phase separation that can be modeled by the phase-separation model mentioned above.

L26: from the experiments. –> from the experiments. This is clear because, to improve this it requires explicit inclusion of both the curvature and twist of the channel with full control over these geometric properties in the model equations (Pudasaini et al., 2005, 2008; Fischer et al., 2012) that has been included here implicitly through the

three-dimensional flow simulations.

L27: by the no-slip boundary condition, –> by the no-slip boundary condition, that could be improved by applying the automatically evolving pressure- and rate-dependent Coulomb-viscoplastic mechanical basal slip conditions developed by Domnik et al. (2013).

P10:

L10-11: I: Such discrepancy could be reduced with phase-separation model.

L25: Also discuss phase-separation effects that might dominate the flow dynamics.

L29: I: This clearly demands for two- or, multi-phase flow model with phase-separation mechanisms.

P11:

L3: model that can be applied to a wide range of debris-flow simulations. –> model that can be applied to a wide range of debris-flow simulations. However, a one or two parameter model, although simple and cheap, may not certainly capture all the more complex debris flows where complexities arise from different material or mechanical parameters and dynamically and locally evolving flow quantities (including the solid, or fluid fraction, and phase velocities, etc.).

L6-11: The new model overcomes a weak point of debris flow modeling: These statements are not fully valid. This discussion should be compatible with Part I. It would be better not to state 'overcomes a weak point' but in practical applications 'just reduces the complicities'. Drag is an essential component of mixture flows. To simplify the situation, and also depending on the flow type, it could be considered to be negligible. Except in local regions, globally flows are essentially thin that can be very economically simulated with real two-phase models that also includes drag (Mergili et al., 2017). So, such descriptions on drag do not help so much in the MS.

[Figure]

L10: difficult to quantify –> difficult to quantify. However, Pudasaini (2012) developed a generalized drag model that overcomes these difficulties. Real complex flows cannot always be modeled by just applying largely oversimplified models. These are different modeling approaches.

L12-14: Not fully true, see other works mentioned above.

L20: The Conclusion needs to be improved accordingly.

P13:

References: Enhance accordingly.

Fig.1: What is 'isometric' here? Mark points for Fig. 4, etc.

Fig. 2: Explain in caption what is: $nu^2$.

Fig. 3: Caption: Reformulate, difficult to understand.

Fig. 4: Write (a), (b), (c) in Fig. panels and improve text accordingly. Also write 'w = 27%', etc. in panels [remove from Caption]. If possible put measurement errors. Mention data source. Also mention at which channel positions.

experiments with water contents of 27% (top), 28.5% (center) and 30% (bottom) –> experiments with different water contents.

What is 'box-averaged'?

Fig. 5: Put legend for: - Exp. depo. margin; - Sim depo. margin.

Write water contents in panels and remove from caption. Put actual coordinates. For w = 28.5% sim. fits very well with exp. Probably explain how such a (almost) perfect fit could be obtained. What is the reason for lower and higher w values simulated inundation/deposition areas are globally wider than experiments? Does this mean there are some special/critical aspects in model and simulations such that it behaves in some typical/critical way with some particular choice of w and the simulation produces less

and less accurate results as you deviate away from this value of w?

Fig. 6: A full 3D photo of the channel would be helpful. Put marks showing positions of measurements. This would help to have an idea on how the super-elevation or the normal load would develop that later effects the viscosity/flow.

Fig. 7: Indicate flow direction, inner-, outer-curvature.

Mention in appropriate place in text: Although only one set of comparison with experiment is shown, mainly with front position, given the complexity of the problem and its intrinsic consequences, such results are important to highlight the potential use of the simulation model to geometrically more complex flows as mostly in literature (except in Pudasaini et al., 2005; 2008) modeling/validations are limited to simple straight or curved and lower dimensional channels.

If possible, also put velocity and viscosity fields; put the flow in red-brown and material-free area in white.

Fig. 9: Fig. description and caption: not clear.

The simulation only covers a single surge runout with homogeneous (unsorted) material.: This would require real two-phase flow model with phase-separation mechanism.

Fig. 10; Fig. 11; Fig. 13: Remove, not needed! Also Fig. 9?

Fig. 14: Explain what generates 'surface wave fronts' and how?

Fig. 15: Why 5 lines 4 legends? Put 32 m, 66 m, etc., on panels. The reason for using SD is not clear. This fig. is not enough discussed in text.
* * *

---

## Author Comment (AC1) · 21 Aug 2017

Please note a .pdf version in the attachment with improved readability.

We thank the referees for the comments, which allowed us to improve the manuscript substantially. The following final authors response is repeating the interactive comment, where R#1 denotes Referee G. Chambon and R#2 stands for comments given by the second referee. We then provide our comment indicated as AC, followed by the new text inserted into the manuscript. We skip cases were we just took over the

suggestions given, for better readability.

R#1: Overall comments

R#1: The choice of the case studies appears a bit questionable, since none of them really challenges the 3D character of the model. A case of impact against an obstacle would have probably been better suited for that purpose.

AC: The scope of the MS is to illustrate the potential of the 3-D model for cases that are typically addressed with depth-averaged approaches. In the case of impact against obstacles, we recommend to include the physics of the coarser colliding grains, which is part of the solver extension presented at EGU 2017. We will address impacts to obstacles in a separate work. It is difficult to find well-documented test cases with detailed measurements for the presented solver, so we limited the range of possible test cases to setups were all details were available to us.

R#1: A more thorough discussion of the respective roles played by the slurry and granular contributions would have been interesting.

AC: We agree, and we see the need for testing the simulation model with further experiments under the viewpoint of distinguishing the roles played by the granular and viscous rheology. However, as both the slurry and the granular rheology model include shear-thinning effects, it is difficult to come up with cases that show the contributions of each rheology separately and still illustrate the model applicability for real-world problems within one paper. However, we tried to address the role played by the granular rheology: we modeled an experiment of the USGS flume with enhanced roughness on the channel bed and compared the simulations of two different granular rheology parameters to illustrate the influence of the granular rheology on basal pressure, flow depth and front flow velocity. We then showed how the model performs in case of a granular dominated mixture with low slurry content, but on a smooth channel bed, which limits the influence of the pressure. To follow the given suggestions, we included a new discussion part addressing the influence of the pressure on the model (see the

inserted text within the detailed comments section):

"4.5: Contribution of the Coulomb-viscoplastic gravel representation within the flow process"

The Herschel-Bulkley representation of the fine material suspension and the representation of the coarser grains as a Coulomb-viscoplastic fluid both introduce shear-thinning behavior to the flow process. An important role of the modeled gravel is the local viscosity increase as a consequence to a local pressure increase. Local pressure variations, as they may occur for example due to roughness elements, lead to a corresponding viscosity variation in the presence of modeled gravel, which in-turn leads to a footprint of the pressure in the shear-rate distribution of the flow field, affecting both the slurry and the gravel rheology. Another consequence of the pressure-dependent rheology model of the gravel is the increase of the viscosity near the bed with increasing flow depth, which enhances the formation of steep flow fronts in the model. Consequently, the front flow depth development within the first half second of laser measurement is well captured in the large scale experiment (see last figure in the manuscript, however, the diagram at (b) lacks grain-size sorting effects). The pressure contribution affects the dam-break release, and one may recognize in Fig. 1 given below that the model is capable of representing the process. Although the front is steeper and arrives earlier due to neglecting the gate, the modeled maximum flow depth reaches the magnitude of the experiments, indicating a realistic material mobilization at least within the first two seconds.

(see Fig. 1: Flow depth over time of the sand-gravel mixture 2 m below the release gate.)

R#1: Detailed comments

R#1: I would personally consider sand as belonging to the granular phase. However, as the authors systematically refer to this granular phase as "gravel", one is led to think that sand is instead accounted for in the slurry. This issue would need to be better

explained and discussed.

AC: Sand-clay suspensions with water have been successfully modeled as shear-thinning viscoplastic materials by several authors in the past. Especially, O'Brian (1988)1 derived from experiments "...that with increasing sand concentration, the viscosity remains comparable to that given from the clay content alone until the sand concentration exceeds about 20% by volume." Sosio et al. (2009)2 found that "The addition of sand finer than 0.106 mm has a negligible effect on it [viscosity] [...] Suspensions with up to 10% of sand have viscosities slightly higher than those composed of the fine fraction alone [...] The relationship between viscosity and total solid concentration deviates from the exponential dependency for sand particles larger than 0.300 mm and sand percentages larger than 35%." Our Herschel-Bulkley rheology model for the slurry is parameterized on the basis of Yu et al. (2013)3 where the material compositions were composed of clay minerals, fine sand, coarse sand, and a small amount (less than 5%) of gravel having diameters less than 10 mm. Yu et al. (2013)3 introduced discontinuities in their modeled yield stress in dependency to the volumetric solid concentration and clay concentration. We added the following statement to the manuscript:

"To be in agreement with the experiments of Yu et al. (2013)3, we consider all particles below 2 mm grain size as part of the interstitial slurry."

R#1: It would be interesting to fully explicit the computation of the lumped rheology in at least some of the examples treated: i.e., give values of the "full" yield stress of the slurry \tau_y (and not only of \tau_00); give values of the effective viscosities of the slurry and granular phases, and of the lumped material (concentration average), for representative shear rates and pressures. More generally, a discussion on the interest of considering this composite constitutive law in the examples shown would be interesting. Is the contribution of the granular phase significant? Would have it been possible to obtain equivalent results with only the viscoplastic part?

AC: We now address the suggested topic in the new discussion part addressing the influence of the pressure on the model.

"3.4: Contribution of the Coulomb-viscoplastic gravel representation within the flow process

The approach of a bulk-averaged viscosity derived from a Herschel-Bulkley representation of the fine material suspension and a Coulomb-viscoplastic representation of the gravel is based on the main assumption that the interstitial fluid can damp the grain collisions up to a degree where the tangential friction between gravel grains dominates the dissipation of the gravel phase. As an example at the limit of applicability, we have chosen a USGS flume experiment that applied a sand-gravel water mixture, in which the collision forces in general cannot be neglected. However, the selected experiment was conducted on a smooth channel bed, such that grain collisions were less pronounced and the video documentation shows a relatively dense material front where the grains are embedded in a slurry within the front. The experiment from the 21st of April 1994 is documented in Major (1997) with a release volume of 9.2 mˆ3, a dry bulk density at release between 1630 and 1960 kg/mˆ3 and a maximal runout length of 16.7 m. For the flow process in the channel, ensemble-averaged data for 11 such smooth-channel SG mixture experiments is available (Iverson et al., 2010). The average wet bulk density at release is 2070 kg / mˆ3 and the water volume within the release body averages 3.17 mˆ3 for the smooth-channel SG experiments (Iverson et al., 2010). We simulate the experiment by representing the sand suspension with the Herschel-Bulkley rheology with 16% water content according to the average numbers for the SG smooth bed experiments (Iverson et al., 2010, table 2 and 3). The gravel is covered by the Coulomb-viscoplastic rheology. We used the same simulation grid as for the SGM smooth bed simulation and applied the same \tau_00 value. The resulting Herschel-Bulkley rheology for the sand suspension had a density-normalized yield stress of \tau_y = 0.0526 mˆ2/sˆ2 and a corresponding consistency factor of k = 0.0158 mˆ2/sˆ2.34. The Herschel-Bulkley exponent was chosen as n = 0.34. The

volumetric share of sand suspension in the mixture was 57.5% and the gravel covers 42.5%. From the integration of the Herschel-Bulkley viscosity over the material volume at the moment of front arrival at the laser at position 32 m, we obtained a volume-averaged slurry viscosity of 14 Pa s that contributes with 57.5% to the overall viscosity. The corresponding volume-averaged gravel viscosity was 54 Pa s, contributing 42.5% to the overall viscosity. The modeled volume-averaged flow process 3.6 seconds after release was thus clearly dominated by the Coulomb-viscoplastic rheology. In a second step, we removed approximately half of the simulated material at 3.6 s after release by excluding cells with pressures over 1500 Pa, which led to a volume-averaged slurry viscosity of 21 Pa s and a Coulomb-viscoplastic average viscosity of 49 Pa s. Thus, the modeled material mixture was dominated by the gravel rheology even within the 50% of material that moved under lower pressures than the rest of the material. An adequate simulation of the experiment as achieved here (Fig. 12 e, f) is not possible only using a Herschel-Bulkley rheology with the parameters linked to the material as in (von Boetticher et al., 2016)"

R#1: I did not really understand the rationale behind equation (1) used to evolve the parameter \tau_00 with water content. In principle, one would expect this parameter to remain constant for a given composition of the solid material, and thus independent of water content. If I understood well, the authors do nevertheless consider a variation of \tau_00 with water content due to a sensitivity of their computations, in particular shear rate, to grid size. From my point of view, this issue should be discussed in more details (see also comment 4 below), and equation (1) should be better justified. In some sense, it can seem disappointing to develop a full 3D model relying on supposedly physically-based constitutive models and, in the end, to use such a trick to resolve what seems to be a purely numerical issue. In particular, the grid-size-sensitivity probably implies that the vertical structure of the flow is relatively poorly captured in the presented simulations. What is then the benefit of the 3D model compared to a depth-averaged approach?

AC: As pointed out in the companion paper, "When simulating laboratory flume experiments where debris-flow material accelerated in a relatively narrow and short channel (Scheidl et al., 2013), a cell height of 1.5 mm, which is of the order of the laboratory rheometer gap, was still not fine enough to reach the limit of grid sensitivity." The grid-size-sensitivity is mainly a consequence of thin layers of high shear as they may appear in debris flows and is less dependent on the overall vertical structure of the flow. However, the correct location, extent and temporal variation of such shear bands is one of the benefits a 3D model can provide compared to a depth-averaged approach. The Herschel-Bulkley rheology of the slurry provides the shear stress as the sum of a shear-rate dependent term and a yield stress. The yield stress is estimated based on material composition due to Yu et al. (2013)[3] and the shear-rate dependent term is a linear function of the yield stress. However, the shear-rate dependent term is a non-linear function of the shear-rate and thus sensitive to the grid discretization and the flow characteristics. Our calibration parameter $\tau\_00$ is not precisely adequate to a physically based parameter because it embodies a countermeasure for grid resolution issues. Although it is multiplied with the function of the volumetric concentrations of clay P0 and solids Cv defined in Yu et al. (2013)[3] to form the Herschel-Bulkley yield stress, the adjustment of $\tau\_00$ should as well overcome the overall disagreement between modeled and real shear stresses for a given calibration case and grid resolution. Even a model that would perfectly adapt to a change in material composition would still face the problem that the change in viscosity due to the new material composition affects the shear rate of the flow which in turn leads to a grid resolution sensitive amplification of the change in shear stress. In a first step, we thus enhance the modeled effect of material composition changes by applying the same relative change to the model parameter.

Please note that the transfer of $\tau\_00$ with equation 1 for the simulation of the 30% water content experiment in the previous manuscript was inconsistent with the other experiments by applying a clay concentration P0 by mass, not by volume. The published model uses the volumetric clay content P0 as suggested by Yu et al. (2013) and
we updated the simulation in the current manuscript accordingly. The corresponding adjustment of the model to changed water contents is still underestimating the water content sensitivity and we see that further research is necessary to find improved modifications of \tau_00 in dependency to changes of the material composition.

R#1: Nothing is said concerning the mesh characteristics used in the different examples presented: grid size, number of elements in the horizontal and vertical directions, etc.

AC: We altered the manuscript and now provide the number of cells in horizontal and vertical directions:

"3.2 Grid resolutions

In general, we distinguished between channelized flows and flows on a plane in choosing our grid resolutions. We first defined a necessary resolution in the flow direction and transverse to the flow to capture the channel geometry. In case of channel flows, we then considered the surface velocity gradients at characteristic front flow velocities in the flow direction and transverse to the flow direction. We kept the ratio between cell length and cell width smaller than the ratio between these longitudinal and transversal velocity gradients and smaller than ten. In case of flows on a runout plane, we kept the ratio below 2.5. The vertical grid resolution was then defined by the available computational resources in a way that results were obtained within reasonable time. The mesh size used for the water content sensitivity experiments increased from 1 mm cell height at the bottom to 4 mm cell height at a distance 25 mm above the bed. This height of 25 mm corresponds to the maximal surface elevation reached at the position of the laser measurement situated one meter downslope of the gate. The cell width was constant 1 cm and the cell length was 2.3 cm. The curved channel experiment was modeled with 39 cells in radial direction and a radial grading from 1 mm cell height and 2 mm cell width at the bed to 3 mm cell height and 2 mm cell width 6.5 cm above the bed. In flow direction, the resolution was constant with a cell length of 5 mm. The USGS flume

with a smooth channel bed had approximately 4 million cells to model the channel flow, which led to a constant cell length of 28 cm, a cell width of 3.3 cm and a grading cell height from 0.7 cm at the bottom to about 1 cm cell height at 19 cm above the bed, which is the highest point reached by the free surface at the laser 32 m downslope of the gate. The runout was modeled with 10 million cells with the same vertical resolution and 5 cm cell sizes in x and y directions. The USGS flume with bumps was represented with 6.5 million cells on a refined mesh, resulting in 1.5 cm cell length, 1.25 cm cell width and 1.4 cm cell height at the bottom. Three cell layers above the bottom the mesh coarsened in lateral and bed-normal direction to 4.4 cm cell length and 2.1 cm cell height. At a height of 32 cm normal to the bed, the mesh coarsened again in the horizontal direction and was continuously graded vertically; however, the corresponding cells were in the air phase of the flow except for the release body. During release, the upper part of the material lies within the coarse mesh, but during column collapse as the flow accelerates, this material transits into the finer mesh closer to the bed, where it starts shearing. We performed a grid resolution sensitivity analysis with the modeled experiments of the water content sensitivity study, as described below."

R#1: The influence of grid size on the presented comparisons with experimental results would also need to be discussed, in particular, whether the results presented in Fig. 5 for the reduced and increased water contents (compared to the calibration case) could be improved with a finer mesh. Same question for the results presented in Fig. 15, notably the strong unphysical oscillations displayed by the pressure signal.

AC: The grid resolution of the USGS flume needed to be fine enough to capture the basal roughness, and we had to simplify the shape of the bumps on the bed because we reached the limit of applicability due to high computational times, which makes the setup unsuitable for grid resolution studies. We see the strong pressure oscillations in Fig. 15 as a consequence to the simplified shape of the basal roughness elements (pyramids instead of cones). We carried out a grid sensitivity analysis for the experiments shown in Fig. 5 and included the findings in the altered manuscript, corrected a

mistake in the composition of the 30% water content experiment simulation (see previous section) and changed the displayed deposits in Fig.5. (In the original MS we showed the wetted slope of the simulations as deposit shape, now we show the material surface defined as the region with a modeled air concentration of 0.5):

"We simulated the three different water content experiments using a coarser grid resolution with twice the cell length, width and height, and conducted the same simulations on a finer mesh that reduced the cell width, length and height by one third. The reduced numerical costs of the coarser mesh allowed running the simulations once without recalibration and once with a recalibration of the experiment with 28.5% water content to the new coarse mesh, followed by adjusted coarse-mesh simulations of the 27% and 30% water content experiments using equation (1). We did not perform a recalibration with the refined mesh due to numerical costs. We only simulated the experiments on the finer mesh applying the original calibration parameter. Table 1 lists a comparison of the resulting runout distances.

[revised manuscript text omitted]

(Literature: 1: O'Brian: Laboratory analysis of mudflow properties, Journal of Hydraulic Engineering, Vol. 114, No 8, 1988. 2: R. Sosio, G. B. Crosta: Rheology of concentrated granular suspensions and possible implications for debris flow modeling, Water Resources Research, DOI: 10.1029/2008WR00692, 2009. 3: Yu, B., Ma, Y., and Qi, X.: Experimental Study on the Influence of Clay Minerals on the Yield Stress of Debris Flows, J. Hydraul. Eng., 139, 364–373, 2013. 4: Z. Wang, J. Yang, and F. Stern: Comparison of Particle Level Set and CLSVOF Methods for Interfacial Flows, 46th AIAA Aerospace Sciences Meeting and Exhibit, Aerospace Sciences Meetings, DOI: 10.2514/6.2008-530, 2008)

Technical comments by R#1:

AC: In general, we changed the manuscript according to the reviewer suggestions. In the following we comment on changes that demand explanation. Comments include the page number P. and line number l. of the original manuscript.

R#1: P.3, l. 25: "Therefore, for each material composition there should be a critical

range where a minor variation in water content causes a strong change in flow depths and run-out distance." I do not really understand this statement. How is this "critical range" related to the exponential variation of the yield stress with water content?

AC: We changed the text to: "Therefore, a minor variation in water content may cause a strong change in flow depths and run-out distance" removing the statement about a range that has a specific high water content sensitivity, because such a range is defined by many factors.

R#1: P.6, l.11-12. Please also indicate the experimental values of tan(\beta_min) and of the corresponding correction factor.

AC: We changed the text to "…which fits the experimental average of $\tan(\beta)$ = 0.33±0.05 (Scheidl et al. (2015) table 2) and the corresponding correction factor $k^*$= 2.1±0.6." The experimental $\tan(\beta)$ is given as a best fit straight line derived from the three laser measurement points and we do not have the resources to construct the set of experimental values $\tan(\beta\_min)$ for all tests. Looking at a representative experiment "Test A 6040_2" in Fig. 2 shows that the inner Laser No 1 does not register any material at the time of the maximum surface super elevation (Fig. 2 below), and a corresponding experimental $\tan(\beta\_min)$ would be 0.37.

(Please see Fig. 2: Example of measured laser time series in a curved channel experiment with 20° slope inclination and mixture A.)

R#1: P.7, l.5-7. Why were simulations of the SG mixture based on the same calibration parameters as for the SGM mixture? Since the composition of these two materials is different, it seems that a recalibration would be necessary? Furthermore, results obtained with SG mixture are not really described in the following (and Fig. 8 is never properly discussed). What is then the point of introducing this additional case?

AC: The case is introduced to show an application of the model were the composition does not contain enough fine material to damp granular collisions, but the absence

of channel roughness seems to allow a viscoplastic approach to model the flow. We now use this additional case as main contribution to illustrate the role of the Coulomb-viscoplastic gravel representation in the newly introduced section. The calibration for the smooth channel bed was done for the SGM experiment introduced in the previous section, based on the front arrival of the corresponding test of 26st of may 1994 published in Major (1997), and as the simulation grid does not change, the model can approximately adjust to the new mixture without recalibration. However, we changed from \tau_00 = 82.78 which was by mistake taken over from the SGM simulation on the rough channel to \tau_00 = 90 as in the smooth channel SGM simulation.

R#1: P.10, l.21-23: "The measured and simulated values do not agree with the mean arrival times implied by the laser signal at position 66 m (Fig. 15 b), however, they do by means of basal pressures for the lower gravel friction angle simulation (Fig. 15 d)." Unclear sentence.

AC: We reformulated the sentence accordingly:

"The measured and simulated values do not agree in terms of the mean arrival times implied by the laser signal at position 66 m (Fig. 12 b) however, when we use the basal pressure signal as an indicator of the front arrival, the measured and simulated arrival times fit well in case of the lower gravel friction angle simulation (Fig. 12 d)."

R#1: P.11, l.1. I do not fully understand what the authors mean by "Our approach allows the model parameters to be linked to (. . .) local topography".

AC: We made the statement more clear:

"Our approach allows the model parameters to be linked to material properties and the model accounts for effects of the local topography on the shear stresses within the material."

R#1: P.12, l.1: "The model can account for the sensitivity of the rheology to channel geometry . . .". This is a strange statement: one does not expect the rheology (a

material property) to be sensitive to channel geometry.

AC: We now state:

"The model can account for the pressure and shear-rate dependent viscous stresses and thereby captures the sensitivity of the material behavior to channel geometry."

R#1: P.12, l.8-10: "Because such changes in model setup are translated into consequences for the flow physics by the model, the ensemble of such simulations may mirror how the modeled site would respond to similar changes." Unclear sentence.

AC: We try to make the statement more clear:

"Because such changes in model setup are translated into consequences for the flow physics by the model, the ensemble of such simulations could be used to outline the consequences of changes at the site. For example, a change in topography by a construction, a change in expected water content by a drainage or a change in expected debris flow compositions by a new gravel deposit could be addressed with the model to visualize the corresponding changes in expected debris flows."

Comments on figures: We followed the given suggestions.

AC: The following comments by referee #2 include the page and line numbers of the original MS, the comments refer to.

R#2: Overall comments

R#2: As the present MS is the application of Part I, the Introduction/Discussion should briefly mention the need of the full 3D simulations, modeling assumptions, simplicity for application as well as the scopes/limitations of the modeling and simulation approaches as mentioned in Part I. This would help the audience who may only focus on application, to directly follow this paper

AC: We added a statement in the introduction pointing out the potential of 3D simulations in comparison to depth-averaged approaches:

"In contrast to the common depth-averaged model approaches for debris flow simulation, this model resolves the flow process in three dimensions. Thus the strong coupling between the flow behavior and the channel geometry and basal roughness can be addressed as shown within this work."

AC: The revised manuscript has a first section were we briefly summarize key assumptions and the approach of the model as well as its restriction to high contents of fine material:

"The model, as described by von Boetticher et al. (2016), is based on an adaptation of the interMixingFoam solver of the open source finite volume code OpenFOAM. We linked the Herschel-Bulkley rheology parameters to the composition of the material mixture and assumed that high contents of fine material such as the interstitial suspension between the gravel grains can damp grain-to-grain collisions Under this assumption, the gravel can be treated as a Coulomb-viscoplastic fluid with the pressure-dependent rheology model of Domnik et al. (2013). The stable implementation together with the reduction to two free model parameters allows reliable numerical studies of three-dimensional flow processes of debris flows that have high shares of fine material. The bulk mixture is combined with an air phase by the Volume-of-Fluid method (Hirt 1981) to capture the free surface In addition to determining typical material parameters (density, water content and relative amounts of gravel and clay), the user is required to input the clay composition (e.g., the fractions of kaolinite and chlorite, illite, montmorillonite; (Yu et al. 2013), and \delta, the friction angle of the gravel fraction, approximated as its angle of repose. To be in agreement with the experiments of Yu et al. (2013), we consider all particles below 2 mm grain size as part of the interstitial slurry. The two remaining calibration parameters are related to the fine material suspension. One of the two free model parameters, the Herschel-Bulkley exponent n, was kept constant and set to 0.34, which was suitable for all simulations presented here. Due to that, the only parameter modified for calibration was tau_00, which acts as a multiplication factor for the calculated yield stress of the fine sediment suspension. In case of dense mixtures where the volumetric solid concentration exceeds a threshold of 0.47, the model amplifies tau_00 as defined in Yu et al. (2013)."

R#2: Writing could be substantially improved in concept and content.

AC: The concept of the paper was to illustrate the model sensitivity to water content, channel geometry and channel bed roughness, and the content followed that concept by presenting the corresponding selected experiments and their simulation. We now make the concept more clear by stating:

"The objective of this study is to illustrate the model's ability to accurately account for a wide range of flow behaviors without recalibration. The key attributes of the model are its sensitivity to water content, gravel- and clay-fraction and clay-mineralogy on the one hand (also see de Haas (2015)), and the interaction between the phase-averaged bulk rheology of the mixture and the complex three-dimensional flow structure on the other. We first present validation test cases that focus on water content sensitivity in laboratory scale, followed by a model setup to analyze the effect of enhanced free surface elevations due to channel curvature. We then study the model's capability to adapt to basal roughness using large-scale flume experiments. Finally, we illustrate the role of the gravel rheology on the overall simulation results using large-scale experiments with a water-sand-gravel mixture. We discuss limitations of the model set-up based on these simulation results."

R#2: Some important dynamical aspects observed in the simulations would have been explained in a better way with elaboration.

AC: In accordance with the comments given by reviewer #1, we now present the role of the coulomb-viscoplastic rheology in an own section (see Authors comment to reviewer #1).

R#2: In a debris flow body, water contain may evolve strongly (Pudasaini and Fischer, 2016; Mergili et al., 2017), and the characteristic may range from dense to dilute flows.

[Figure]

These aspects need to be clearly mentioned in the MS. Recent and relevant literatures could be included and discussed.

AC: We agree and included the suggested literature.

R#2: Detailed comments

AC: We agree with most of the comments and only list here the changes that went beyond the suggested improvements or explain why we did not follow the suggestions.

R#2: Abs.: material properties were known –> material properties and compositions were known AC: As we list the compositions within the following enumeration, we think it is clear that the composition is seen a property of the debris flow mixture.

R#2: (including its mineral composition): Remove

AC: The presented work is the first debris flow model that accounts for the clay mineral composition, therefore we should mention this within the abstract. The necessary model ingredients are now mentioned as:

"For the selected experiments in this study, all necessary material properties were known – the content of sand, clay (including its mineral composition) and gravel as well as the water content and an angle of repose of the gravel. Given these properties together with the density of the mixture, two model parameters are sufficient for calibration, and a range of experiments with different material compositions can be reproduced by the model without recalibration."

R#2: two model parameters are sufficient for calibration –> two model parameters are used for calibration

AC: We cannot change the statement that way because we use only one parameter for calibration, however, from previous discussions we know we should mention that two parameters are available for calibration.

R#2: The angle of repose: is this 'the angle of repose' of 'internal friction angle'?

AC: The internal friction angle is difficult to measure for coarse gravel, whereas it is simple to estimate an angle of repose from the material deposits in the field. Whenever available, we used the angle of repose as a measure for the granular friction. However, we used the internal friction angle as an alternative approximation in the large scale experiments.

R#2: P2: L24: a restricted multiplication factor: Explain.

AC: We changed the statement to:

"Due to that, the only parameter modified for calibration was \tau_00, which acts as a multiplication factor for the calculated yield stress of the fine sediment suspension. In case of dense mixtures where the volumetric solid concentration exceeds a threshold of 0.47, the model amplifies \tau_00 as defined in (Yu et al. 2013)."

R#2: L33: We mention that super-elevation has been analytically modeled and validated for dry granular flows and flows of mixtures by Pudasaini et al. (2005, NHESS; 2008, PoF)

AC: We added the reference to the literature, but instead of summarizing the general discussion content we point out a difference to this study by naming its depth-averaged approach:

"We mention that effects of curvature were analytically modeled and validated for dry granular flows and flows of mixtures by (Pudasaini et al. 2005, Pudasaini et al. 2008) with a depth-averaged approach."

R#2: P3: L21: Since, in the mixture, largely the solid particles exhibit slip, viscous fluid exhibits no-slip along the basal surface such distinct basal boundary conditions can only be included with real two-phase mass flow models (see, e.g., Pudasaini, 2012; Mergili et al., 2017).

AC: We included the statement as:

"In general, largely the solid particles exhibit slip and viscous fluid exhibits no-slip along the basal surface. Only through the core assumption of identical velocity between gravel and surrounding fluid due to high drag do such distinct basal boundary conditions reduce to a no-slip condition. OpenFOAM offers partial-slip boundary conditions, however, the definition would only become meaningful together with real two-phase mass flow models as in Pudasaini (2012); Mergili et al., (2017), or as developed by Wardle (2013) within our 3D framework."

R#2: P4: The mixture effectively consists of water, fine particles and gravel with different physical parameters and mechanical, hydrodynamical response to applied loads. In such a complex situation, how a simplified model with two free parameters can capture the flow so nicely. It needs to be discussed. There is an almost perfect fit between the shapes of the experimental and simulated deposits in the calibration case (Figure 5 center).: Explain the reliability of the perfectness, because the other two panels do not so strongly support this statement.

AC: As can be seen from the comparison of the modeled flow depth with laser data, the depth of the flow along the material body is well captured by the model in general. As we calibrate the model to fit the maximal runout and thus capture the longitudinal profile is well, the overall width of the deposit matches the experiment more-or less as a consequence of volume conservation. The local width of the deposit is adjusting the transversal surface gradient to balance the hydrostatic pressure at the center line. Again, as the hydrostatic pressure along the center line is well captured, the width of the deposit performs well, too. However, we had an inconsistent application of the share of clay in terms of mass and in terms of volume in case of the 30% water content experiment, and the model adjustment to water content changes is now less exact but more consistent (the sensitivity is underestimated for both decreasing and increasing water contents). But by adjusting \tau_00, an adequate fit can be achieved for other water contents, too: Fig. 3 shows the final deposit for the 30% water content experiment with \tau_00 recalibrated to 24 Pa.

(Please see Fig. 3: Measured deposit (red line) and simulation of the 30% water content experiment after recalibrating the free model parameter to 24 Pa match the maximal runout.)

R#2: L6-11: Improve.

AC: We simplified the text to be more clear:

"An initial and simple approach chosen here is to first evaluate the relative effect that a changed water content has on the modeled Herschel-Bulkley yield stress \tau_y. In a second step, we apply the same relative change to the calibration parameter \tau_00. Let \tau_newycal be the Herschel-Bulkley yield stress calculated by the model for a new water content, based on the original value of the calibration parameter \tau_00cal that is not yet adjusted to the new water content. We reduced or increased the free model parameter \tau_00 according to equation 1, where \tau_00_cal denotes the Herschel-Bulkley yield stress as calculated by the model in the calibration test before the water content changed."

R#2: Even if the flow height is twice the maximum grain size what about the experimental simulation reliability/reproducibility/accuracy?

AC: We added a grid sensitivity analysis section, as commented in the reply to reviewer #1. We repeated the simulation of the 28.5 water content on three different clusters in the ETH domain (Hera, Brutus, Euler) and on a Centos machine as well as on a Ubuntu laptop without noting any difference in results.

R#2: P5: L1-2: However, the simulated front also temporarily paused at x = 2.04 m, until it was overrun by a second wave 0.1 s later: Not shown.

AC: We removed that statement.

R#2: L3-4: The maximum flow depth and the subsequent decrease are well reproduced (Figure 4 center): There is no flow depth here; in Fig. 4 hydrograph, in Fig. 5 only deposition areas are shown. L3-9: There is a non-logical switching between Fig.

4 and Fig. 5; difficult to follow.

AC: The figures show flow depth over time, not a hydrograph.

R#2: L12: However, super elevation also occurs in dry granular flows as this phenomenon is primarily induced by the geometry of the channel (curvature and twist) rather than the viscous or frictional properties of the material.

AC: We consider enhanced super elevation as highly dependent on frictional properties.

R#2: L12: so it can be viewed as a further indicator for model quality. –> so it can be viewed as a further indicator for model quality (from geometric point of view).

AC: The enhanced surface elevation is the result of a flow process and has the same potential for quantitative comparison of models with experiments as front position over time or flow depth.

R#2: P6: L6: For this a mechanical phase-separation model (Pudasaini and Fischer, 2016) would be required.

AC: We added:

"To account for the granular flow front, a mechanical phase-separation model like that described by (Pudasaini and Fischer, 2016) would be required, or even a coupled Lagrangian particle simulation."

R#2: L14: the front volume: not clear/not seen. L21-22: Does it upscale?

AC: We could not yet remodel the experiment to larger scales.

R#2: P7: L18:Separation between solid- and fluid-type materials may lead to this discrepancy that can be described with phase-separation model (see literature mentioned above).

AC: Although the separation in a granular front and a viscous tail may be described with

phase-separation models, the surges addressed here, to our understanding, originate from the reservoir. We tracked such surges in the video documentation from the reservoir down, in the three experiments as shown in Fig. 10 of the new manuscript. With a smooth channel bed as discussed at P7 L18 in the original MS, the corresponding grainsize sorting is even less pronounced.

R#2: L15-16: This discrepancy could have been emerged due to the fact that there could be substantial interactions and also separation between solid- and fluid-type phases that has not been considered in the simulations.

AC: We tend to agree with Iverson (2010) and see the reason for different front arrival times between basal pressure and laser measurements in single grains jumping ahead that are only captured by the laser.

R#2: P9: L7-8: Only grid-resolution (numerical) is explained as a possible source of discrepancy. But this discrepancy could also be reduced by applying real two-phase models with explicit phase-interactions. Needs discussion.

AC: We added a grid sensitivity analysis section to the text and rewrote the discussion of the water content sensitivity experiments accordingly:

[revised manuscript text omitted]

R#2: L20: As mentioned previously other relevant works could be discussed. L21: Discuss the work by de Haas et al. (2015) and for grain sorting and phase separation that can be modeled by the phase-separation model mentioned above.

AC: We had a long review of previous debris flow models in the first paper version and a discussion section of models with comparable approaches in the last paper version, but as a result of the open discussion of the previous articles, we focus on the abilities and drawbacks of this model presented here and do not discuss other model approaches, especially as the text is already long.

R#2: L26: from the experiments. –> from the experiments. This is clear because, to improve this it requires explicit inclusion of both the curvature and twist of the channel with full control over these geometric properties in the model equations (Pudasaini et al., 2005, 2008; Fischer et al., 2012) that has been included here implicitly through the three-dimensional flow simulations.

R#2: We do not agree, as we consider the three dimensional finite volume or finite element approaches superior in accuracy to depth averaged approaches. The difference in volume originates from the simplified release box, were the original transition to a half-pipe channel caused the material to jump and include air bubbles, something that

can be captured with appropriate high grid resolution but cannot be captured by the models mentioned by the reviewer.

R#2: L27: by the no-slip boundary condition, –> by the no-slip boundary condition, that could be improved by applying the automatically evolving pressure- and rate-dependent Coulomb-viscoplastic mechanical basal slip conditions developed by Domnik et al. (2013).

AC: We added:

"One could reduce the amount of material sticking to the walls by applying partial slip conditions like in Domnik et al. (2013), however, this would demand a multiphase approach to account for the wetting of the walls, which goes beyond the scope of this model."

R#2: P10: L10-11: I: Such discrepancy could be reduced with phase-separation model. L25: Also discuss phase-separation effects that might dominate the flow dynamics. L29: I: This clearly demands for two- or, multi-phase flow model with phase-separation mechanisms.

AC: We now briefly mention the possible approaches to account for phase separation:

"Phase separation effects would need to be taken into account by implementing either drift-flux models, multiphase approaches with one Navier-Stokes equation per phase or coupled Lagrangian particles or coupled discrete element methods. However, the corresponding model extension would introduce new model parameters and higher numerical costs. As a consequence of our reduced approach without grainsize sorting effects, we did not model the run-out patterns of the rough channel experiments, in contrast to the smooth channel experiment where less demixing occurred."

R#2: P11: L6-11: The new model overcomes a weak point of debris flow modeling: These statements are not fully valid. This discussion should be compatible with Part I. It would be better not to state 'overcomes a weak point' but in practical applications

'just reduces the complicities'. Drag is an essential component of mixture flows. To simplify the situation, and also depending on the flow type, it could be considered to be negligible. Except in local regions, globally flows are essentially thin that can be very economically simulated with real two-phase models that also includes drag (Mergili et al., 2017). So, such descriptions on drag do not help so much in the MS. L10: difficult to quantify –> difficult to quantify. However, Pudasaini (2012) developed a generalized drag model that overcomes these difficulties. Real complex flows cannot always be modeled by just applying largely oversimplified models. These are different modeling approaches.

AC: We now name it "reduces the complexities" instead of "overcomes a weak point". The corresponding section is inspired by the feedback of applicants in the debris flow protection domain and is in fact the main motivation for the presented work.

R#2: L12-14: Not fully true, see other works mentioned above.

AC: As long as there is no published work modeling such variety of flows, mixtures and scales with one parameter, we consider our statement as true.

Comments on figures: We followed the given suggestions were possible, exceptions are discussed below.

R#2: Fig. 13: Remove

AC: The figure explains the inhomogeneous time development of the flow front positions in the previous figure, which is the core figure of the paper and consumed the largest efford and computational costs within this work. We therefore think it is a key contribution to the paper.

R#2: Fig. 14: Explain what generates 'surface wave fronts' and how?

AC: We think that the discussion about the generation of surface wave fronts is beyond the scope of this paper.

R#2: Fig. 15: Why 5 lines, 4 legends?

AC: In analogy to (Iverson 2010) we plot the standard deviation which is a grey line above and below the measured data

Please also note the supplement to this comment:
https://www.geosci-model-dev-discuss.net/gmd-2017-25/gmd-2017-25-AC1-supplement.pdf

[Figure]

**USGS Flume Smooth Bed
SG Experiments**

h [m] at pos. 2m

Figure showing h[m] at pos. 2m versus Time [s], with curves: Average SG experiments, Standard deviation, Simulation.

**Fig. 1.**

**Test A 6040_2**

Surface Elevation [mm]

Time [s]

Laser 1

Laser 2

Laser 3

**Fig. 2.**

[Figure]

**Fig. 3.**